# Dynamics of parafermionic states in transport measurements

Ida E. Nielsen[1,2]*, Jens Schulenborg[1,3], Reinhold Egger[4] and Michele Burrello[1,2]

**1** Center for Quantum Devices, Niels Bohr Institute, University of Copenhagen, DK–2100 Copenhagen, Denmark
**2** Niels Bohr International Academy, University of Copenhagen, DK–2100 Copenhagen, Denmark
**3** Department of Microtechnology and Nanoscience (MC2), Chalmers University of Technology, S–412 96 Göteborg, Sweden
**4** Institut für Theoretische Physik, Heinrich Heine Universität, D–40225 Düsseldorf, Germany
* ida.nielsen@nbi.ku.dk

May 17, 2023

## Abstract

**Advances in hybrid fractional quantum Hall (FQH)-superconductor platforms pave the way for realisation of parafermionic modes. We analyse signatures of these non-abelian anyons in transport measurements across devices with $\mathbb{Z}_6$ parafermions (PFs) coupled to an external electrode. Simulating the dynamics of these open systems by a stochastic quantum jump method, we show that a current readout over sufficiently long times constitutes a projective measurement of the fractional charge shared by two PFs. Interaction of these topological modes with the FQH environment, however, may cause poisoning events affecting this degree of freedom which we model by jump operators that describe incoherent coupling of PFs with FQH edge modes. We analyse how this gives rise to a characteristic three-level telegraph noise in the current, constituting a very strong signature of PFs. We discuss also other forms of poisoning and noise caused by interaction with fractional quasiparticles in the bulk of the Hall system. We conclude our work with an analysis of four-PF devices, in particular on how the PF fusion algebra can be observed in electrical transport experiments.**

## 1   Introduction

Experimental advances in the field of fractional quantum Hall (FQH) systems and their integration with superconducting elements [1, 2] have sparked hope to engineer parafermionic zero-energy modes. These modes, for brevity also called parafermions (PFs), are emergent topological low-energy excitations that obey non-abelian braiding statistics. This makes them an interesting building block for engineering phases of matter with exotic kinds of topological order and potential candidates to encode quantum information in a non-local way [3]. PFs are predicted to be localised at domain walls between e.g. a superconductor (SC) and a ferromagnet, each inducing a gap in the same counterpropagating fractional edge modes [4–9]. Promising signals of the required superconducting coupling and the related crossed Andreev reflection have been reported in Ref. [2]. Here a superconducting thin finger of niobium nitride was positioned in the trench of a graphene Hall bar encapsulated in boron nitride, and a negative Hall resistance was measured for several FQH states including the incompressible liquid with filling $\nu = 1/3$. In what follows, we focus on this specific filling factor.

It has been argued that the effects of dissipation via single-electron tunnelling into SC vortices may play an important role for these experimental observations [10], which therefore constitute a promising but not conclusive indication of the onset of PFs. Alternative and more direct experimental investigations of hybrid FQH-SC systems are thus desirable to verify the appearance of these fractional low-energy modes. To this purpose, Ref. [11] proposed FQH-SC setups to study the physics of PFs based on electronic transport measurements. The simplest scenario is offered by devices hosting a pair of these non-abelian anyons, the state of which can be characterised by their shared number of fractional ($e/3$) electron charges modulo one Cooper pair. When coupling them to a normal metallic lead, their fractional charge mod $1e$ and thus their fusion channel can be detected indirectly by transport spectroscopy: the conductance reveals the energy spectrum of the two-PF system which, in turn, depends on their conserved charge $\tilde{q}e/3$, where $\tilde{q} = 0, 1, 2$ is the reduced charge number.

The ideal topological protection of this degree of freedom is broken if fractional quasiparticles enter or leave the PF system and change the charge by $\pm e/3$. Such quasiparticle poisoning could for example originate from a coupling with the gapless edge modes of the FQH liquid. If the related poisoning rate is weak with respect to all other energy scales of the system, the onset of a three-level telegraph noise in the conductance signal is expected [11]. In this work, we build a framework to study the features of the transport dynamics and noise of these PF systems out of equilibrium. We distinguish the case in which PFs are incoherently poisoned by a fractional quasiparticle bath from the case of PFs coherently coupled to fractional parti-

cles in the system as, for instance, in the presence of antidots formed in the bulk of the FQH liquid [12, 13]. In both cases the shared factional charge $\tilde{q}$ is not conserved, but for coherent couplings, the state of the PFs must be described, in general, by a linear superposition of the different charge states. We will adopt a quantum jump method [14, 15] to model how the continuous current readout in a two-PF device turns into a projective measurement of this degree of freedom for sufficiently long times. The same method will allow us to study the onset of a telegraph noise present both when including incoherent sources of fractional particles and when the coherent dynamics of the system competes with the continuous current measurement.

We will present the limitations and time scales over which the fusion of pairs of PFs can be measured in systems with more than two PFs. This is crucial for engineering protocols for the manipulation and measurement of larger sets of PFs with the potential applications towards topological quantum computation in mind. In particular, both the investigation of the associativity of the anyonic fusion rules, and the braiding, must be based on the possibility of manipulating (at least) four PFs and measuring their fusion outcomes (see for instance Refs. [16, 17]). We will propose a proof-of-principle protocol to detect the associativity rules restricted to the fractional charge $\tilde{q}$ through transport measurements and estimate its limitations in terms of time scales as dictated by the out-of-equilibrium quantum jump simulations.

The structure of this work is as follows: In Sec. 2 we present the basic setup to study the dynamics of PFs coupled to an external electrode and introduce in Sec. 2.1 the quantum jump method to describe the evolution of this open system. We discuss in Sec. 2.2 the measurement induced projection of superpositions of states into separate charge sectors and give estimates for the current within each of these in Sec. 2.3. In Sec. 3 we include a coupling to the external edges of the FQH system and use the same quantum jump technique to model poisoning by including jump operators associated with fractional quasiparticles. We find that the current displays three-level telegraph noise corresponding to $e/3$ charges entering or leaving the PF system. In Sec. 4 we consider an antidot as a coherent poisoning source and discuss the resulting two-level telegraph noise. In Sec. 5 we extend the description to a system of four PFs and investigate their fusion algebra and a possible protocol for studying its associativity rules in Sec. 5.1. In Sec. 5.2 we propose a construction to realise tunability of the PF couplings. We present our conclusions in Sec. 6. The appendices provide technical details of the quantum jump method (App. A), the extension of jump operators to fractional modes (App. B), numerical results for strong environment interaction (App. C) and details on the associativity matrix of $\mathbb{Z}_6$ PFs (App. D).

We set Planck's reduced constant as well as the Boltzmann constant to one in this work unless explicitly stated: $\hbar = k_{\mathrm{B}} = 1$.

## 2 Readout dynamics of two parafermions

A minimal setup for studying transport readouts of PFs is sketched in Fig. 1. A thin SC is placed in a trench of the $\nu = 1/3$ FQH liquid (a graphene, boron nitride and graphite heterostructure in Ref. [2]) and induces superconducting pairing between counterpropagating edge modes in the presence of strong spin orbit coupling (as in niobium nitride). At the ends of the SC the edge modes are backscattered by the quantum Hall gap and, from a low-energy field theory description of the system, one $\mathbb{Z}_6$ PF is expected to emerge at each end of the SC [5,7]. Each PF is described by an effective operator $\alpha_i$, such that $\alpha_1 \alpha_2 = \mathrm{e}^{i\pi/3} \alpha_2 \alpha_1$. One of the PFs is coupled to a normal metallic lead at a voltage bias $V_b$ with respect to the grounded SC. The state of the two PFs can be characterised by a parity operator $\mathrm{e}^{-i\pi/6} \alpha_2^\dagger \alpha_1 = \mathrm{e}^{-i\pi q/3}$ [18] where the number operator $q$ counts the fractional charges $e/3$ modulo one Cooper pair in the segment

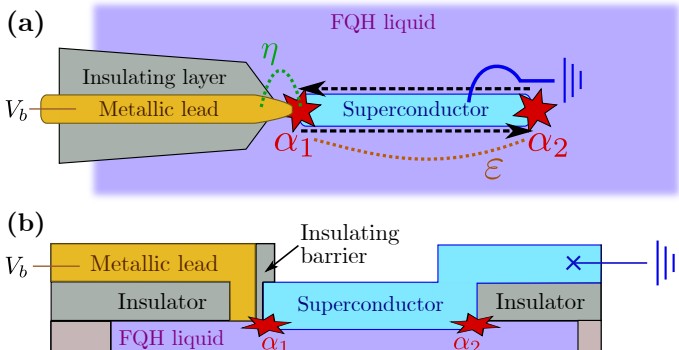

Figure 1: **(a)** Top and **(b)** section view of a two-PF device with a grounded SC (light blue) inducing pairing between the chiral edge modes [black dashed arrows in **(a)**]. At each end of the SC there is a localised PF (red stars) and the two couple with a strength $\varepsilon$. The metallic lead (gold) is coupled only to the left PF $\alpha_1$ and isolated from the SC and the edges of the FQH liquid (purple) by insulating layers (grey).

of edge modes gapped by the SC. Analogously to Majorana modes in a topological SC, the two PFs interact via a coupling $\varepsilon$, exponentially suppressed with their distance. Consequently, the sixfold degenerate groundstate of the system is split by

$$H_{2\mathrm{pf}} = -2\varepsilon \cos(\pi q/3 + \phi), \qquad (1)$$

where the phase $\phi$ depends on the chemical potential of the edge modes and the length of the SC [19,20]. For a single pair of PFs this splitting, and thereby $q$ (mod 3), can be read indirectly by a conductance measurement where a current runs between the lead and the grounded SC, through the PFs.

To discuss this readout procedure, we describe the coupling with the lead by the Hamiltonian $H_c = i\eta\alpha_1^3\left(l + l^\dagger\right)$, where $\eta$ is the coupling strength between the lead and the left PF, and $l$ is the annihilation operator for an electron at the end of the lead [21–25]. With this coupling, $q$ is no longer a conserved quantity, but the six PF states are divided into three sectors labelled by the fractional charge $\tilde{q} = q \bmod 3$, which is a degree of freedom protected against the electron tunnelling. The Hamiltonian thus separates into three independent non-interacting models. At resonance for a given $\tilde{q}$ sector, $eV_b = 4\varepsilon \cos(4\pi\tilde{q}/3 + \phi) \equiv \Delta_\varepsilon(\tilde{q})$, the zero-temperature conductance is quantized to $G_{\tilde{q}} = 2e^2/h$. In general, the conductance can assume three different values depending on $\tilde{q}$ for suitable $\eta$, $V_b$, $\varepsilon$ and temperature $T$. These values can be resolved as long as the phase $\phi$ results in sufficiently different energy splittings for each $\tilde{q}$ sector [11]. In this work we present data for the optimal point $\phi = \arctan(1/\sqrt{27})$, but our results do not qualitatively depend on this specific choice.

Based on the results in Ref. [19], the estimates of the PF energy splitting in Ref. [11] and realistic parameters in hybrid graphene FQH-SC devices [2], it is convenient to introduce the energy scale $\lambda = 2\pi \cdot 2.4$ GHz, which we will refer to throughout this work. Unless otherwise specified we take $\varepsilon = \lambda$ and $T = \lambda/3$.

## 2.1 Electron jump operators for a two-parafermion system

To obtain reliable predictions that can be tested in future experiments, we analyse the dynamics of an open system in which the PFs are coupled to the external environment provided by the metallic electrode. In this work, we focus on the electrical current signatures of such effects. To study the coupling with the metallic lead, we assume that this electron reservoir behaves as a Markovian bath, i.e. the correlation time of electrons in the lead is the shortest time scale

in the system dynamics [26]. An ideal tool to calculate the evolution of such systems is the stochastic quantum jump method [14, 15, 27–29]. With that, we can calculate trajectories for different initial states of the PF system and estimate the evolution of the fractional charge $\tilde{q}$ and the current through the normal lead. To this end, we introduce jump operators $L_+^e$ and $L_-^e$ that move a charge $e$ from the lead to the PFs and vice versa. Following [26, 29] we define them as

$$L_\pm^e = i \sum_{q=0}^{5} \sqrt{\Gamma J_\pm(\Xi_q, V_b)} |q\rangle \langle q + 3|, \quad \Gamma = 2\pi \nu_l |\eta|^2, \tag{2}$$

where the coupling rate $\Gamma$ depends on the density of states in the lead $\nu_l$, and $J_{+(-)}$ is the two-point correlation function of the $l(l^\dagger)$ operator evaluated at the energy $\Xi_q = 4\varepsilon \cos(\pi q/3 + \phi)$. This energy describes transitions between eigenstates with charge numbers $q$ and $q \pm 3$ of $H_{2\text{pf}}$ in Eq. (1). In terms of the Fermi-Dirac distribution, $n_{\text{F}}(\omega, V_b) = (1 + \exp[(\omega - eV_b)/T])^{-1}$, the correlation function reads:

$$J_\pm(\Xi_q, V_b) = n_{\text{F}}(-\Xi_q, \pm V_b). \tag{3}$$

Our definition of the jump operators relies on an assumption of immediate projection of electrons in the lead, meaning that the application of $L_\pm^e$ directly corresponds to a charge moving back or forth. This allows us to obtain an estimate of the current by keeping track of the number of times the two jump operators are applied under the evolution of the PF state. We quantify the current signal by averaging the difference of electron jumps in and out of the PF system over a time window $t_{\text{avg}}$. The trajectories of the quantum jump method are calculated with Monte Carlo simulations where the state evolution is divided into small time steps $\delta t$ with respect to $2\pi \Gamma^{-1}$. Within each time step, there is a small probability $\propto \Gamma \delta t$ that one of the jump operators is applied to the state thereby changing the PF charge $q \to q \pm 3$. Further details of the protocol are described in App. A, and the source code is available online [30]. We do not include the Lamb shift in the treatment of the PF systems since there are no near-degeneracies in $H_{2\text{pf}}$ (avoiding $\phi$ close to multiples of $\pi/6$), and it only gives an overall shift of the spectrum and small corrections $\lesssim 3\nu_l |\eta|^2 \approx \Gamma/2$ [26]. We stress the advantage of the quantum jump technique for the purpose of calculating distinct trajectories, relevant for comparing with experimental outcomes, whereas the Lindblad master equation [26] would provide only an ensemble average over the individual trajectory realisations.

## 2.2 Projection of charge sectors

With the method outlined above, we investigate now the time evolution of a two-PF system described by the Hamiltonian $H_{2\text{pf}}$ in Eq. (1). We focus on the reduced charge $\tilde{q}$ and investigate how it is projectively measured by the current. We fix the coupling rate to $\Gamma = 0.1\lambda$ from here on.

If the initial state $|\psi_i\rangle$ is an eigenstate of $\tilde{q}$, i.e. a linear superposition of the states $|q\rangle$ and $|q + 3 \mod 6\rangle$, the time evolved state remains within the same charge sector. If instead the initial state is a superposition of states with different $\tilde{q}$, the current measurement modelled by the jump operators projects to a fixed, but random $\tilde{q}$ for sufficiently long times. To test a "worst-case" scenario, we take as the initial state an equal superposition of all the charge states $|\text{equal}\rangle = \frac{1}{\sqrt{6}} \sum_{q=0}^{5} |q\rangle$ with an initial expectation value of the reduced charge number $\langle \tilde{q} \rangle = 1$. Every jump of an electron between the lead and the PFs consists a weakly projective measurement and on average, $\langle \tilde{q} \rangle$ should with equal probability $1/3$ evolve into 0 or 2 or remain 1. Fig. 2(a) depicts the evolution of $\langle \tilde{q} \rangle$ (colour scale) in a single random trajectory for different values of the voltage bias $V_b$. We observe that on a timescale of order $2\pi \Gamma^{-1} \approx 4$ ns, the interaction with the lead is not sufficient to project the state into a given $\tilde{q}$ sector except from a few cases for voltages around the resonance energies $\Xi_q$. Inspired by the inverse participation

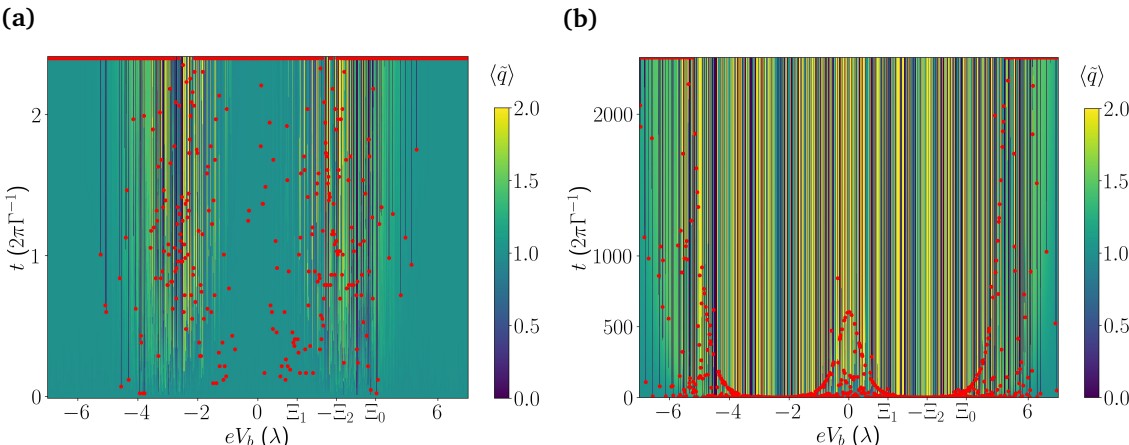

Figure 2: Evolution of $\langle \tilde{q} \rangle$ at different $V_b$ for the initial state |equal⟩ of a two-PF system characterized by $\varepsilon = \lambda = 2\pi \cdot 2.4$ GHz, $T = \lambda/3$ and $\Gamma = \lambda/10$. Red dots mark $\zeta^* = 0.9$ for each trajectory and a dot in the very top is to indicate that this threshold is not reached within the simulation time. **(a)** On the time scale $\sim O(2\pi\Gamma^{-1})$ few of the trajectories reach the threshold $\zeta^* = 0.9$. **(b)** Differently, the majority of trajectories reach this threshold on the $O(10^3(2\pi\Gamma^{-1}))$ timescale.

ratio, we introduce the following quantitative measure of the projection of a general state $|\psi\rangle = \sum_{q=0}^{5} c_q |q\rangle$:

$$\zeta = \sum_{\tilde{q}=0,1,2} \left( |c_{\tilde{q}}|^2 + |c_{\tilde{q}+3}|^2 \right)^2 . \tag{4}$$

This is 1/3 for the initial state |equal⟩ and evolves towards 1 when the state is progressively projected into only one of the charge sectors. In Fig. 2(a) we mark by a red dot the time $t_{\text{proj}}$ when $\zeta$ on a trajectory reaches the value $\zeta^* = 0.9$. The many red dots in the top are to illustrate that our projection measure has not reached this threshold within the simulation time. This supports our interpretation of the uneven distribution among charge sectors as a consequence of the fact that the state is not projected on this time scale and remains a superposition of $\tilde{q}$ sectors.

On a much longer time scale of $O(10^3(2\pi\Gamma^{-1}))$, we see better the projection of states (Fig. 2(b)). Still, the projection is slowest at voltages away from $\Xi_q$ according to both $\langle \tilde{q} \rangle$ and our $\zeta^*$ measure. In additional simulations that we do not show here, both estimates indicate that the projection time is shortened with an increase in temperature as one would expect. We remark that the projection time is highly sensitive to the initial state and that it decreases considerably for the simpler superposition of charge sectors $\frac{1}{\sqrt{3}}(|q=0\rangle + |q=1\rangle + |q=2\rangle)$, as shown in App. A. Furthermore, the projection times depicted in Fig. 2(b) show that the current measurements should be performed at voltages around the $\Xi_q$ resonances in order to minimize the readout time of the charge $\tilde{q}$. The qualitative behaviour of the projection time as a function of $V_b$ and $T$ can be also estimated by analysis of the decay rates of the wave function components which characterise the continuous part of the evolution during a stochastic trajectory. In App. A we argue on this basis that, as mentioned above, the state |equal⟩ represents a worst-case scenario in terms of projection time for the purpose of a readout of $\tilde{q}$.

## 2.3  Current behaviour of a two-parafermion system

As described in Sec. 2.1, we can obtain an estimate of the current trajectories in the metallic lead connected to the PFs with the quantum jump method. Here we present predictions for the

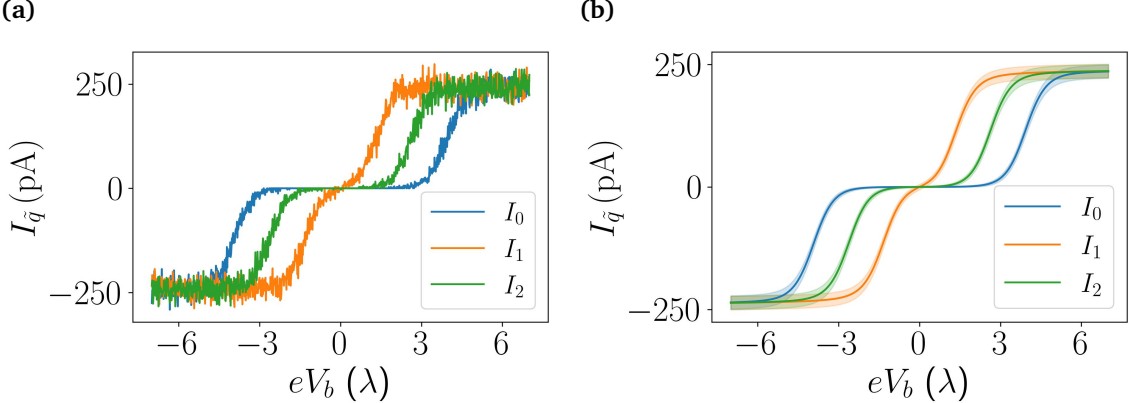

Figure 3: Current estimates $I_{\tilde{q}}$ in the three different charge sectors $\tilde{q} = 0, 1, 2$. **(a)** $I_{\tilde{q}}$ obtained with stochastic jump simulations of single trajectories. The current is averaged over a time $t_{\mathrm{avg}} = 0.1\,\mu s \approx 24(2\pi\Gamma^{-1})$, beginning at $t > 2000(2\pi\Gamma^{-1})$. The initial states are eigenstates of the conserved charge $\tilde{q}$ and no transitions between different sectors can occur. **(b)** Analytical prediction from the Landauer-Büttiker formalism for the current and 1 sigma noise (shaded width). We use the same value $\delta\nu^{-1} = 0.1\,\mu s$ for the time averaging interval as in **(a)**. For the cutoff $\Omega$ we use $20\lambda$.

current signal including noise for experimental situations where the reduced charge number $\tilde{q}$ is a conserved quantity. As a benchmark of our method, we compare our results for the average current and noise to the ones of the Landauer-Büttiker formalism [31, 32].

In Fig. 3(a) we plot estimates of the current within the three different charge sectors $\tilde{q} = 0, 1, 2$: for different values of the bias $V_b$, we initialise the PFs in an equal superposition of $|q = \tilde{q}\rangle$ and $|q = \tilde{q} + 3\rangle$ and consider a single random trajectory as dictated by the quantum jump method. For each $\tilde{q}$, the PF state remains a superposition of $|\tilde{q}\rangle$ and $|\tilde{q} + 3\rangle$. At the different voltages, we report the average value of the current $I_{\tilde{q}}$ numerically observed during a time window $t_{\mathrm{avg}} = 0.1\,\mu s$ (see App. A for further details). From Fig. 3(a) the current signals of the different sectors appear to be distinguishable for certain ranges of $V_b$.

We now compare this with the non-perturbative result of the Landauer-Büttiker formalism, applicable for effectively non-interacting systems like the two-PF setup treated in this section. We state below the zero-temperature differential conductance, the finite temperature current and the zero-frequency noise, respectively:

$$G_{\tilde{q}}(\omega) = \frac{2e^2}{h} \frac{(2\omega\Gamma)^2}{(\omega^2 - \Delta_\varepsilon^2(\tilde{q}))^2 + (2\omega\Gamma)^2}, \tag{5}$$

$$I_{\tilde{q}}(V_b) = \int_0^{V_b} dV \int_{-\Omega}^{\Omega} d\omega\, G_{\tilde{q}}(eV) \left( \frac{-dn_{\mathrm{F}}(\omega, V)}{d\omega} \right), \tag{6}$$

$$S(0) = \int_{-\Omega}^{\Omega} d\omega\, \frac{G_{\tilde{q}}(\omega)}{\delta\nu} \Big\{ (2\omega\Gamma)^2 \big[ n_{\mathrm{F}}(\omega, V_b) n_{\mathrm{F}}(-\omega, -V_b) + n_{\mathrm{F}}(\omega, 0) n_{\mathrm{F}}(-\omega, 0) \big]$$
$$+ (\omega^2 - \Delta_\varepsilon^2(\tilde{q}))^2 \big[ n_{\mathrm{F}}(\omega, V_b) + n_{\mathrm{F}}(\omega, 0) - 2n_{\mathrm{F}}(\omega, V_b) n_{\mathrm{F}}(\omega, 0) \big] \Big\}. \tag{7}$$

Here $\Omega$ is a large ultraviolet cutoff frequency and $\delta\nu^{-1}$ is a time interval over which the result is averaged. As mentioned above, $\Delta_\varepsilon(\tilde{q}) = 4\varepsilon\cos(4\pi\tilde{q}/3 + \phi)$ is the energy splitting within the $\tilde{q}$ sector. In Fig. 3(b) we plot these expressions for current and noise with the same parameters as in Fig. 3(a) and one sees that they agree well with the numerical estimates. From these results we conclude that it is possible to distinguish the signals from each sector for appropriate values of the voltage bias.

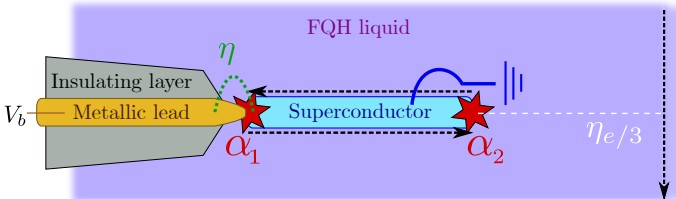

Figure 4: Sketch of quasiparticle poisoning due to coupling with external edge modes. The SC and PFs are in the bulk of the FQH system whereas the source of fractional charges is assumed to be the outermost edges of the FQH liquid.

The results discussed so far rely on the conservation of the charge $\tilde{q}$ in the open two-PF system. Two important aspects, however, must be further considered. First, a realistic experimental scenario will be characterized by fractional quasiparticle poisoning. Such poisoning is for instance responsible for the onset of telegraph noise in FQH Fabry-Perrot interferometers [33] (see also the related simulations in Ref. [34]). In order to successfully perform a transport readout of the PF state, it is necessary that any poisoning time is much larger than the projection times estimated in this section. Second, to obtain a linear superposition of states with different charge $\tilde{q}$, it is necessary to include in the analysis of the PF setup a further subsystem that can coherently exchange fractional quasiparticles with the PFs. In the following sections, we address these important points.

## 3  Quasiparticle poisoning and three-level telegraph noise

The setup presented in the previous section is inspired by the hybrid FQH-SC systems analysed in Refs. [1,2] (see also the theoretical setups in Ref. [35]). These platforms can in general be subject to two sources of fractional quasiparticle poisoning that affect the stability of the fractional charge $\tilde{q}$. The first is given by quasiholes and quasiparticles in the bulk of the incompressible Hall liquid. Since these fractional excitations are characterised by an energy gap $\Delta_{\text{FQH}}$, it is expected that they become localised in a glass state by disorder effects at low temperatures [34]. This scenario is consistent with the interferometry experiments in GaAs platforms (see, for instance, Refs. [33,36,37]), and suggests that bulk excitations provide only a minor contribution to poisoning events for $T \ll \Delta_{\text{FQH}}$. The second source is more relevant and is constituted by the chiral gapless edge modes in the system. To include this source of poisoning in our analysis, we consider the external edge of the system as a reservoir of fractional quasiparticles, and we assume a weak incoherent coupling between the chiral gapless modes and one of the two PFs. This is illustrated in Fig. 4. Specifically, we assume the coupling to be

$$H_{c,\text{edge}} = \eta_{e/3}(\alpha_2 \psi_{e/3}^\dagger + \psi_{e/3}\alpha_2^\dagger), \tag{8}$$

where $\psi_{e/3}$ is the edge mode annihilation operator taken at a given position along the edge, and only the right PF ($\alpha_2$) couples to this gapless mode. More general interactions coupling both PFs with an extended region of the edge can be considered and do not qualitatively affect our results. We expect the tunnelling amplitude $\eta_{e/3}$ of the fractional quasiparticles to decrease with the distance of $\alpha_2$ from the edge and with the energy gap $\Delta_{\text{FQH}}$.

Based on the coupling in Eq. (8), we now introduce two additional jump operators $L_{\pm}^{e/3}$ describing incoherent jumps of $e/3$ charges from the edge into the PF system and vice versa. These operators are derived in App. B by extending the procedure in Refs. [26, 29] to the case of fractional charge operators. They depend on the anyonic spectral function $\tilde{d}$ that results from the Fourier transform of the two-point correlation function in time of the operator

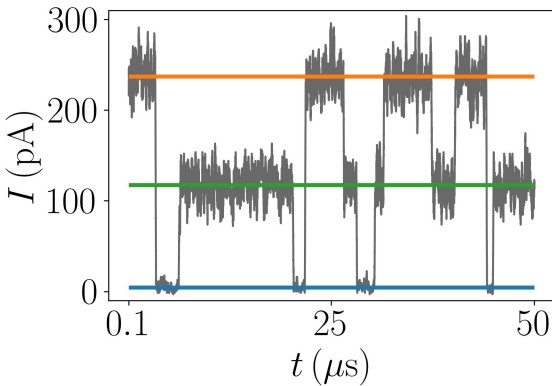

Figure 5: Current evolution at fixed $eV_b = 2.6\lambda$ with a coupling strength to the edges $\eta_{e/3} = 3.5 \cdot 10^{-3}\lambda$. The initial state is $|\text{equal}\rangle$, but on the displayed timescale the projection happens very fast, $O(10\text{ns})$, at the specific voltage. The blue, orange and green horizontal lines correspond to the mean value of the current in the sectors $\tilde{q} = 0, 1$ and $2$, respectively, in the absence of poisoning. Jumps between the current levels are due to a charge $e/3$ entering or leaving the PF system via the outermost FQH edge. For the FQH gap in graphene we use the typical value $\Delta_{\text{FQH}} = 1.7$ meV [41] and we set $\mu = 0$.

$\psi_{e/3}$. In particular, we consider the explicit expression for this spectral function presented in Refs. [38, 39], and by including $\Delta_{\text{FQH}}$ as an ultraviolet cutoff we obtain:

$$\tilde{d}(E, \mu) \equiv \frac{e^{-\beta(E-\mu)/2}}{2\pi\mathbf{\Gamma}(\frac{1}{3})\Delta_{\text{FHQ}}^{1/3}\beta^{1/3}}\mathbf{\Gamma}\left(\frac{1}{6} + \frac{i\beta(E-\mu)}{2\pi}\right)\mathbf{\Gamma}\left(\frac{1}{6} - \frac{i\beta(E-\mu)}{2\pi}\right). \tag{9}$$

Here $\mu$ is the chemical potential of the external edge mode (corresponding to the potential of the incoming edge mode in Ref. [2]), $\beta$ is the inverse temperature of the system and the bold $\mathbf{\Gamma}$ represents the Euler gamma function (see App. B for further details and Ref. [40] for related results). The resulting expression for the fractional jump operators is

$$L_\pm^{e/3} = \sum_q \sqrt{\Gamma_{e/3}\tilde{d}(E_{q\mp1} - E_q, \pm\mu)}\, e^{i\frac{\pi}{6}(1\mp2q)} |q\rangle \langle q \mp 1| \,, \tag{10}$$

where we introduced the fractional poisoning rate $\Gamma_{e/3} = 2\pi|\eta_{e/3}|^2\beta$, and $E_q$ labels the eigenvalues of $H_{2\text{pf}}$ in Eq. (1). Eq. (10) implies that the average rate of poisoning events is affected by the anyonic distribution $\tilde{d}$. It is important to notice that, as in the case of the Bose-Einstein distribution, $\beta\tilde{d}$ diverges at $E = \mu$ in the zero-temperature limit due to anyonic condensation [38]. Therefore, in the special resonant cases where $E_{q\mp1} - E_q = \pm\mu$, the low-temperature limit of Eq. (10) would give rise to a dramatic increase of the effective poisoning rate, and the assumptions necessary for applying the quantum jump method would no longer be valid. These zero-temperature resonances, however, are on one side easily avoided by small shifts of $\mu$ which can be controlled by varying the potential of the edge. On the other side, they have only a negligible effect at experimentally relevant temperatures $\geq 10$ mK and for a FQH gap $\Delta_{\text{FQH}} = 1.7$ meV (see App. B for further details).

Using the quantum jump method with the two jump operators in Eq. (10) together with the ones in Eq. (2), we can get numerical estimates of the current from the external lead to the system also in the presence of poisoning effects. Again, our results rely on the assumption that the edge modes relax on a short time scale like the lead modes (Markovian approximation). In Fig. 5 we show a stochastic trajectory of the current as a function of time at a fixed voltage for a

weak poisoning rate $\Gamma_{e/3} \approx 0.002\Gamma$ and the initial state $|\text{equal}\rangle$. We observe jumps between the three current levels of $\tilde{q} = 0, 1, 2$ that are otherwise well separated and stable. Furthermore, we recognise that the jumps can happen from one level to any of the other two. For the chosen parameters the poisoning time is $2\pi\Gamma_{e/3}^{-1} = 1.8\,\mu s$ and much longer than the projection time which is $O(10\,\text{ns})$. The example of current evolution in Fig. 5 displays less frequent occupation of the $\tilde{q} = 0$ sector than of the other two. We quantify this and confirm that it is a general trend by considering the Lindblad master equation [26] for the steady state density matrix $\rho_0$:

$$0 = \partial_t \rho_0 = -i\left[H_{\text{pf}}, \rho_0\right] + \sum_{\substack{s=\pm \\ c=e,e/3}} L_s^c \rho_0 (L_s^c)^\dagger - \frac{1}{2}\{(L_s^c)^\dagger L_s^c, \rho_0\}. \tag{11}$$

The steady state obtained with the parameters stated in Fig. 5 yields probabilities $0.17, 0.44$ and $0.39$ for the charge sectors $\tilde{q} = 0, 1$ and $2$, respectively, which fits well the example in Fig. 5. By increasing the fractional rate $\Gamma_{e/3}$ we find that $\Gamma_{e/3} \approx 0.02\Gamma$ is an approximate upper bound for the three-level telegraph behaviour to remain observable assuming a time averaging window of $t_{\text{avg}} = 0.1\,\mu s$. This corresponds to a coupling strength $\eta_{e/3}$ three times the one used in Fig. 5 and we show an example of the related current evolution in App. C. For stronger poisoning rates, it becomes difficult to distinguish the current signal of the $\tilde{q} = 0$ sector from noise. A clear observation of the three-level telegraph noise relies indeed on both a sufficient signal-to-noise ratio, and on the fact that all charge sectors must be populated for sufficiently long times with respect to the poisoning time. An exhaustive analysis of the necessary conditions to observe the three-level telegraph noise is beyond our scope, given the non-trivial dependence of the system dynamics on all the physical parameters. We can, however, identify important qualitative features; in general, the signal integration time and the choice of the bias voltage of the lead are instrumental to achieve a sufficiently large signal-to-noise ratio, whereas the temperature, FQH edge voltage and FQH gap concur in determining the poisoning rates, and thus the occupation of the different charge sectors.

From our numerical results, we conclude that if the poisoning rate $\Gamma_{e/3}$ is weak compared to the measurement rate $\Gamma\left(\Gamma_{e/3} \lesssim 0.02\Gamma\right)$, the current signal will jump between all three levels that correspond to the different charge sectors $\tilde{q}$, given that the voltage is tuned appropriately (see Fig. 3). On a time scale that is long compared to the poisoning time (and thereby also the measurement time) the signal will therefore show a three-level telegraph noise behaviour.

## 4 Coupling with an antidot

Another source of poisoning could be impurities or electric potential hills trapping fractional charges in the bulk of the FQH liquid. These are commonly referred to as antidots (see for instance Ref. [42]). We consider a coherent form of quasiparticle poisoning and include one antidot in a simplified manner as a two-level system [43] described by the operators $(a, a^\dagger)$, fulfilling $a^2 = (a^\dagger)^2 = 0$. The two levels correspond to two different charges on the antidot, $Q_{\text{ad}}$ and $Q_{\text{ad}} + e/3$, split in energy by $\delta_{\text{ad}} \ll \varepsilon, \Gamma$. We require the commutation relations $a\alpha_i = e^{i\pi/3}\alpha_i a$ and $[\alpha_i, Q_{\text{ad}}] = 0$, which reflect the interacting nature of the system. Furthermore, we introduce the number $q_{\text{ad}} = \{0, 1\}$ counting the excess number of fractional charges on the antidot. We take the coupling between the PFs and the antidot to be

$$H_{\text{pf}-\text{ad}} = \sum_{i=1,2}\left(\eta_{\text{ad},i}\alpha_i^\dagger a + \eta_{\text{ad},i}^* a^\dagger \alpha_i\right), \tag{12}$$

where $\eta_{\text{ad},i}$ is the coupling strength between PF $\alpha_i$ and the antidot. In Fig. 6 we sketch this coupling. We can describe the total 2PF+antidot system with a common unitary evolution and

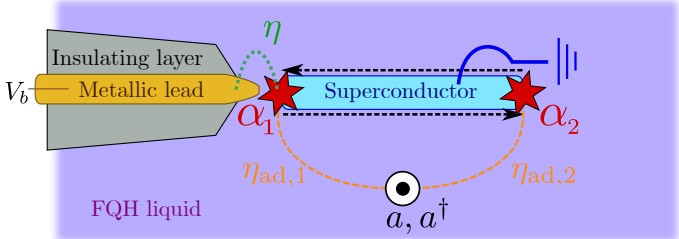

Figure 6: Sketch of an antidot coupling to both PFs. Here we ignore the outermost edges of the FQH system, but include still the normal lead to perform conductance measurements.

take as our basis

$$|q, q_{ad}\rangle = \left(a^\dagger\right)^{q_{ad}} \left(\alpha_1^\dagger\right)^q |0,0\rangle = \{|0,0\rangle, |0,1\rangle, \dots |5,1\rangle\}. \tag{13}$$

Based on this definition and the matrix elements $\langle q' | \alpha_2^{(\dagger)} | q \rangle$ stated in App. B, we find that the elements of the total system Hamiltonian are

$$\begin{aligned}
\langle q', q'_{ad} | H_{tot} | q, q_{ad}\rangle =& \left(-2\varepsilon \cos(\pi q/3 + \phi) + \delta_{ad}\delta_{q_{ad},q'_{ad}}\delta_{q_{ad},1}\right)\delta_{q,q'} \\
&+ \left(\eta_{ad,1} + \eta_{ad,2}\, e^{-i\frac{\pi}{6}(1+2q)}\right)\delta_{q',q+1}\delta_{q'_{ad},0}\delta_{q_{ad},1} \\
&+ \left(\eta_{ad,1}^* + \eta_{ad,2}^*\, e^{-i\frac{\pi}{6}(1-2q)}\right)\delta_{q',q-1}\delta_{q'_{ad},1}\delta_{q_{ad},0}. \tag{14}
\end{aligned}$$

The jump operators for electrons tunneling to or from the normal lead have to be extended to take into account the extra degree of freedom from the antidot. Importantly, the current measurement can still only sense the charge of the PFs and does not change the state of the antidot. In the new basis, the jump operators are then given by

$$\tilde{L}_\pm^e = i \sum_{q, q_{ad}} \sqrt{\Gamma J_\pm(\Xi_q, V_b)} |q, q_{ad}\rangle \langle q+3, q_{ad}|. \tag{15}$$

The dynamics of the PF and antidot system, coupled to the external electrode, still preserves the total fractional charge mod $e$ of the system. The 12 states in Eq. (13) are therefore split into three charge sectors, each including four states that, importantly, are characterised by two different values of the PF charge $\tilde{q}$ due to the possible leakage of charge into the antidot. As a result, the new 12×12 Hamiltonian and the corresponding jump operators yield current signals as those exemplified in Fig. 7. They show telegraph noise behaviour, but now only between two $\tilde{q}$ sectors since the antidot can only host charge $Q_{ad}$ or $Q_{ad} + e/3$. For example, the initial state in Fig. 7(b) is the superposition $\frac{1}{\sqrt{2}}(|0,0\rangle + |3,0\rangle)$ with charge $Q_{ad}$ on the antidot. The current signal in the $\tilde{q} = 0$ sector is around only a few pA for the chosen parameters. However, the state evolves, due to $H_{pf-ad}$, into a superposition of the states $|5,1\rangle$ and $|2,1\rangle$ with one $e/3$ charge moved to the antidot and thus a current signal around 120 pA emerges. The signals are similar for all initial states of the kind $\frac{1}{\sqrt{2}}(|q=\tilde{q},1\rangle + |q=\tilde{q}+3,1\rangle)$. The onset of the two-level telegraph noise is due to the competition between the weak coherent coupling $\eta_{ad,i}$ with the antidot and the stronger incoherent coupling $\Gamma$ with the normal lead (with $\eta_{ad,i} = 0.05\Gamma$ in Fig. 7). Increasing the coupling $\eta_{ad,i}$ the jumps between current levels become more frequent. Once one reaches the regime $\eta_{ad,i} \gtrsim \Gamma$, the currents for the three initial states used in Fig. 7 instead become stable again like in Fig. 7(a), but at different values. We refer to App. C for such a plot. The three new levels correspond to the leading transition within each of the three fractional charge sectors: $(\tilde{q} + q_{ad})$ mod 3.

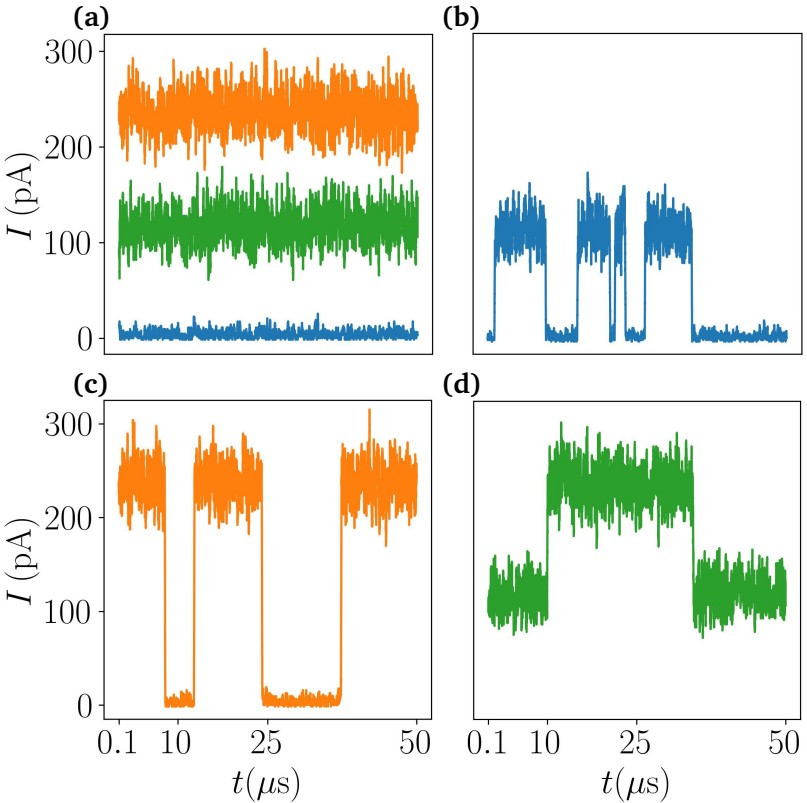

Figure 7: Electrical current signals in the presence of an antidot: Numerically estimated current evolution over times $\gg 2\pi\eta_{\mathrm{ad},i}^{-1} \approx 80$ ns for the three different initial states **(b)** $\frac{1}{\sqrt{2}}(|0,0\rangle + |3,0\rangle)$, **(c)** $\frac{1}{\sqrt{2}}(|1,0\rangle + |4,0\rangle)$ and **(d)** $\frac{1}{\sqrt{2}}(|2,0\rangle + |5,0\rangle)$ with $\delta_{\mathrm{ad}} = 5\cdot10^{-4}\lambda$ and coupling strengths $\eta_{\mathrm{ad},1} = \eta_{\mathrm{ad},2} = 5\cdot10^{-3}\lambda$. In **(a)** we show for comparison the current for the same initial states when $\eta_{\mathrm{ad},i} = 0$. The colours here correspond to those in **(b)-(d)**. As in Fig. 5 we fix $eV_b = 2.6\lambda$.

In conclusion, our simulations suggest that, when the coherent coupling with the antidot is weaker than $\Gamma$, the current measurement provides a reliable estimate of the PF state, provided that the readout occurs over sufficiently short time scales. Importantly, even though $2\pi|\eta_{\mathrm{ad},i}|^{-1} \lesssim t_{\mathrm{proj}} \lesssim t_{\mathrm{avg}}$ for the parameters adopted in the previous calculations (the projection time is inferred from the data in Fig. 2(b) around $eV_b = 2.6\lambda$), we find that less than 2% of the trajectories display jump events between the current levels for times $t < 2\pi|\eta_{\mathrm{ad},i}|^{-1}$.

Given our analysis of quasiparticle poisoning and telegraph noise, which are determined both by the FQH edge and by the coupling with antidots in the bulk, we conclude that the readout of the state of two PFs through the coupling with an external lead can be performed as long as $\Gamma$ is sufficiently larger than $\Gamma_{e/3}$ and $|\eta_{\mathrm{ad},i}|$. In the following, we consider the ideal case of readouts fast enough to avoid the poisoning of the PF charge.

## 5 Four-parafermion devices

Most of the properties of non-abelian anyons require systems with more than two of these quasiparticles to be studied. Therefore we now extend our analysis to four $\mathbb{Z}_6$ PFs. We consider a device with two thin SC islands analogous to the previous proposals such that in total, four PFs $\{\alpha_1, \ldots, \alpha_4\}$ become accessible. Additionally, we assume for simplicity that couplings to

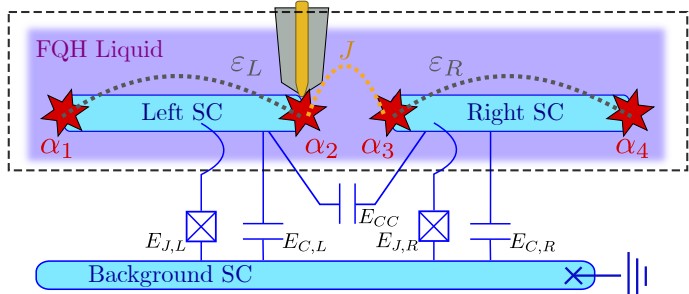

Figure 8: Schematics of a proposed setup to manipulate four PFs. Two SC fingers, each within a trench in the bulk FQH liquid, host in total four PFs. The dashed box highlights the relevant PF couplings whereas the details below are suggested ingredients to construct tunable couplings. Both SCs are coupled to a background grounded SC with a transmon construction of Josephson energy $E_{J,i}$ and charging energy $E_{C,i}$, $i = L, R$. A cross capacitance generating the coupling $E_{CC}$ is considered as well. The low-energy model describing the PFs is characterised by pairwise interactions $\varepsilon_L, \varepsilon_R$ and $J$. At the very top we sketch the metallic lead (gold) that is coupled to $\alpha_2$.

antidots and external edges of the FQH system are negligible.

To manipulate the quantum states of a four-PF system, we must consider a device in which the couplings can, in general, be controlled in time. To this purpose we focus on the device depicted in Fig. 8, closely related to setups discussed in Refs. [43–45]. We present a detailed discussion of this system in Sec. 5.2. For the moment, we focus on the effective PF model sketched in the dashed box in the top of Fig. 8. The couplings between $\alpha_1$ and $\alpha_2$, and between $\alpha_3$ and $\alpha_4$, are of electrostatic origin. We assume that they are tunable and label them with $\varepsilon_L$ and $\varepsilon_R$, respectively. The modes $\alpha_2$ and $\alpha_3$ are instead coupled with a strength $J$, modelling a fractional Josephson interaction [7, 46, 47]. Additionally, we consider a global charging energy interaction $\varepsilon_{LR}$ that depends on the total charge of the system. The corresponding four-PF Hamiltonian reads (see Sec. 5.2 for details)

$$
\begin{aligned}
H_{4\mathrm{pf}} = \Big\{ &-\varepsilon_L\, e^{-i\left(\frac{\pi}{6}+\phi_L\right)} \alpha_2^\dagger \alpha_1 - \varepsilon_R\, e^{-i\left(\frac{\pi}{6}+\phi_R\right)} \alpha_4^\dagger \alpha_3 - J\, e^{-i\left(\frac{\pi}{6}+\theta\right)} \alpha_3^\dagger \alpha_2 \\
&-\varepsilon_{LR}\, e^{-i\left(\frac{\pi}{3}+\phi_L+\phi_R\right)} \alpha_2^\dagger \alpha_1 \alpha_4^\dagger \alpha_3 \Big\} + \mathrm{H.c.}
\end{aligned}
\tag{16}
$$

The phases $\phi_L$, $\phi_R$ play a role analogous to $\phi$ introduced in Sec. 2. $\theta$ is instead determined by the large magnetic field in the device and can be tuned by small variations of this. It yields a similar effect in determining the energy levels associated with the fractional Josephson term. Like in the setups analysed in the previous sections, the fusion of $\alpha_1$ and $\alpha_2$ is characterised by their shared fractional charge which we now label $q_L$ for clarity. Similarly, there exists an operator $q_R$ counting the number of charges shared by $\alpha_3$ and $\alpha_4$. For an isolated system, the total charge mod $2e$ of the four PFs, $(q_L + q_R)e/3 \equiv Qe/3$, must be conserved. Hence, the 36 dimensional Hilbert space of $H_{4\mathrm{pf}}$ splits into six sectors corresponding to different $Qe/3 = 0, e/3, \ldots, 5e/3$, each of six states. The eigenstates of $H_{4\mathrm{pf}}$ can thus be labelled by the total charge and their energy level $E_n^{(Q)}$ within this sector; $\{|Q, n\rangle\}$, $n = 1, \ldots, 6$.

The Hamiltonian in Eq. (16) corresponds to a two-site quantum clock model where the PF operators are related to the $\mathbb{Z}_6$ clock operators $\tau_i$ and $\sigma_i$, $i = L, R$, by a generalised Jordan-Wigner transformation [4]:

$$
\alpha_1 = \sigma_L, \qquad \alpha_2 = e^{i\frac{\pi}{6}} \tau_L \sigma_L, \qquad \alpha_3 = \sigma_R \tau_L, \qquad \alpha_4 = e^{i\frac{\pi}{6}} \tau_R \sigma_R \tau_L. \tag{17}
$$

The unitary clock operators satisfy $\tau_i^6 = \sigma_i^6 = \mathbb{1}$ and the commutation relation $\sigma_i \tau_i = e^{i\frac{\pi}{3}} \tau_i \sigma_i$. In addition to the parity operators $P_L = e^{-i\frac{\pi}{6}} \alpha_2^\dagger \alpha_1 = \tau_L^\dagger = e^{-i\frac{\pi}{3}q_L}$ and $P_R = e^{-i\frac{\pi}{3}q_R}$, we can

introduce the corresponding $P_J = \mathrm{e}^{-i\frac{\pi}{6}} \alpha_3^\dagger \alpha_2 = \sigma_R^\dagger \sigma_L = \mathrm{e}^{-i\frac{\pi}{3} j}$, where $j$ is the dual of $q$ and describes the fusion of $\alpha_2$ with $\alpha_3$. In terms of the clock operators the four-PF Hamiltonian reads

$$H_{4\mathrm{pf}} = \left\{ -\varepsilon_L\, \mathrm{e}^{-i\phi_L}\, \tau_L^\dagger - \varepsilon_R\, \mathrm{e}^{-i\phi_R}\, \tau_R^\dagger - J\, \mathrm{e}^{-i\theta}\, \sigma_R^\dagger \sigma_L - \varepsilon_{LR}\, \mathrm{e}^{-i(\phi_L + \phi_R)}\, \tau_L^\dagger \tau_R^\dagger \right\} + \mathrm{H.c.} \qquad (18)$$

Due to the conservation of the total charge $Q$, we can rewrite $\tau_R^\dagger = \mathrm{e}^{-i\pi Q/3}\, \tau_L$. Furthermore, in the basis $\{|Q, q_L\rangle\}$ the coupling between $\alpha_2$ and $\alpha_3$ acts as:

$$\sigma_R^\dagger \sigma_L\, |Q, q_L\rangle = |Q, q_L - 1\rangle\,, \qquad \sigma_L^\dagger \sigma_R\, |Q, q_L\rangle = |Q, q_L + 1\rangle\,. \qquad (19)$$

Hence, in this basis, $H_{4\mathrm{pf}}$ reads

$$\begin{aligned}
\langle Q', q_L'|H_{4\mathrm{pf}}|Q, q_L\rangle = & \left\{ -2\varepsilon_L \cos(\pi q_L/3 + \phi_L) - 2\varepsilon_R \cos(\pi(Q - q_L)/3 + \phi_R) \right. \\
& \left. - 2\varepsilon_{LR} \cos(\pi Q/3 + \phi_L + \phi_R) \right\} \delta_{Q',Q} \delta_{q_L', q_L} \\
& - J(\mathrm{e}^{i\theta} \delta_{q_L', q_L+1} + \mathrm{e}^{-i\theta} \delta_{q_L', q_L-1}) \delta_{Q',Q}\,.
\end{aligned} \qquad (20)$$

Now we introduce a coupling between $\alpha_2$ and an external metallic lead, $i\eta\alpha_2^3(l + l^\dagger)$. Analogously to the two-PF case, this mixes sectors with different $Q$ causing transitions from $Q$ to $Q \pm 3 \pmod{6}$ with amplitudes proportional to $\langle Q \pm 3, m|\alpha_2^3|Q, n\rangle$, where $m, n$ still label energy eigenstates of $H_{4\mathrm{pf}}$. Thus the Hilbert space is now divided into three sectors of 12 states each and there exist in general 36 possible transitions within each sector. In the basis $\{|Q, n\rangle\}$, we can express the electron jump operators as

$$\hat{L}_\pm^e = \sum_{Q,m,n} \sqrt{\Gamma J_\pm(E_n - E_m)}\, \langle Q + 3, m|\tau_L^3 \sigma_L^3|Q, n\rangle\, |Q + 3, m\rangle\, \langle Q, n|\,, \qquad (21)$$

where $\Gamma$ and $J_\pm$ are as defined in Sec. 2.1 and we rewrote $i\alpha_2^3 = \tau_L^3 \sigma_L^3$. Before analysing the features of the resulting quantum trajectories, let us consider different limits of the PF couplings. Based on that, we suggest a protocol for experimentally investigating the associativity of the fusion rules of the PFs and use the quantum jump technique to make predictions for the related current signatures in Sec. 5.1.

We begin with the limit $J \ll \varepsilon_L, \varepsilon_R$ where the two SCs simply act as copies of the original two-PF setup with each their charge number conserved. The set $\{|q_L, q_R\rangle\}$, which is the eigen-basis of the $\tau$ operators, approximates the eigenstates of the Hamiltonian up to corrections of order $J/\min_{q_L}(E_{Q,q_L} - E_{Q,q_L+1})$, where $E_{Q,q_L} |Q, q_L\rangle = H_{4\mathrm{pf}}(J = 0) |Q, q_L\rangle$. A current readout with the lead coupled to $\alpha_2$ at a strength $\Gamma \gg J$ results in the behaviour reported in Fig. 3(a) with $q_R$ "frozen", analogously to the measurement dynamics discussed in Sec. 4: the coupling with the normal lead competes with the fractional Josephson energy $J$ that may cause jumps between current readouts corresponding to different values of $\tilde{q}_L$. This would manifest itself in a telegraph-noise of the current signal between two or three values depending on the level spacing of the right SC. In special cases where e.g. $\phi_R = -\pi/6$, such that the right SC has degenerate levels for $q_R = 0$ and $q_R = 1$, the limit $J \ll \Gamma \ll \varepsilon_L \ll \varepsilon_R$ translates to the two-PF and antidot case treated in Sec. 4 with respect to the left SC where $J$ corresponds to $\eta_{\mathrm{ad},i}$. If instead $\Gamma \ll J \ll \varepsilon_L, \varepsilon_R$, the tunnelling spectroscopy across the normal lead will reflect the complex energy spectrum of $H_{4\mathrm{pf}}$ and no telegraph noise is observed.

In the limit $\varepsilon_L, \varepsilon_R \ll J$, the charges $q_L, q_R$ are not separately conserved and the eigenstates of the $P_J$ operator, $\{P_J |Q, j\rangle = \mathrm{e}^{-i\frac{\pi}{3} j} |Q, j\rangle\}$, approximate the eigenstates of $H_{4\mathrm{pf}}$ (again, up to corrections of order $\max(\varepsilon_L, \varepsilon_R)/\min_j(E_{Q,j} - E_{Q,j+1})$ with $E_{Q,j} |Q, j\rangle = H_{4\mathrm{pf}}(\varepsilon_L = \varepsilon_R = 0) |Q, j\rangle$). The eigenenergies are grouped into six subsets each containing six almost degenerate levels with splitting of the order $\max(\varepsilon_L, \varepsilon_R)$, and the situation simplifies to the two-PF case for

$\varepsilon_L = \varepsilon_R = 0$ with $\alpha_1$ and $\alpha_4$ isolated. For finite $\varepsilon_L, \varepsilon_R$, a current measurement through $\alpha_2$ with $\varepsilon_L, \varepsilon_R \ll \Gamma \ll J$ is expected to show three-level telegraph noise as in Sec. 3, corresponding to jumps between different $|\tilde{j}\rangle$ states where $\tilde{j}$ is the $\mathbb{Z}_3$ part of $j$, analogously to $\tilde{q}_i$.

Regardless of the limits taken above, the role of $\varepsilon_{LR}$ is to shift the resonance energies (corresponding to $\Delta_\varepsilon(\tilde{q})$ defined in Sec. 2) and thereby also the optimal voltage range for the readouts discussed above.

## 5.1 Associativity relations of the anyonic parafermion fusion

The associativity relations of the algebraic fusion rules of a collection of anyons constitute a crucial element to define their non-abelian character [48, 49]. In the following, we present a protocol to experimentally verify that the PFs in the setup depicted in Fig. 8 are indeed anyons characterised by non-abelian fusion rules. This corresponds to showing that any pair of PFs defines a degree of freedom that can assume 6 different values, and that the degrees of freedom obtained by coupling the same PF with different partners, for instance $\alpha_2$ with $\alpha_1$ and $\alpha_3$, obey specific algebraic rules, and do not depend on the first PF ($\alpha_2$) alone. In particular, given the set of four PFs depicted in Fig. 8, the interactions labelled by $\varepsilon_L$ and $J$ are related to the fusion of $\alpha_2$ with $\alpha_1$ and $\alpha_3$, respectively. The outcomes of these two different fusion processes are represented by the eigenstates of $P_L$ and $P_J$, respectively, and the related associativity relations are described by a unitary transformation, usually called the $F$-matrix, that maps the eigenstates of $P_J$, $|Q, j\rangle$, into the eigenstates of $P_L$, $|Q, q_L\rangle$.

As mentioned in Sec. 2, the outcome of the fusion of two $\mathbb{Z}_6$ PFs (for instance the eigenstates of $P_L$) can be characterized by considering a $\mathbb{Z}_2$ fermionic parity, which is constantly scrambled by the interaction with the normal lead, and the more robust $\mathbb{Z}_3$ fractional charge $\tilde{q}$. This structure is a general mathematical feature, and it holds also for the fusion of the PFs $\alpha_2$ and $\alpha_3$ belonging to different SC islands. The $\mathbb{Z}_6$ fusion algebra is indeed naturally decomposed in these two degrees of freedom. Ref. [5] showed, for instance, that the braiding operations of $\mathbb{Z}_{2m}$ PFs can be decomposed into the tensor product of a $2 \times 2$ Majorana braiding matrix with an $m \times m$ unitary matrix affecting the fractional degree of freedom. In the following, we show that also the associativity $F$-matrix splits into the tensor product of two associativity matrices for the different sectors. To this end, we introduce the rewriting $e^{i\frac{\pi}{3}q_L} = i\alpha_2^3 \alpha_1^3 e^{i\frac{4\pi}{3}\tilde{q}_L}$ [11] and the corresponding notation $|q_L\rangle = |p_L, \tilde{q}_L\rangle$ where $p_L = (1 - i\alpha_2^3\alpha_1^3)/2$. Similarly, $|j\rangle = |p_J, \tilde{j}\rangle = |(1 - i\alpha_3^3\alpha_2^3)/2, \tilde{j}\rangle$. Considering the system in Fig. 8, the $6 \times 6$ PF associativity matrix $F^{(6)}$ can be derived from the mapping into clock degrees of freedom [50],

$$|Q, q_L\rangle = \sum_j F^{(6)}_{q_L, j} |Q, j\rangle = \frac{1}{\sqrt{6}} \sum_{j=0}^{5} e^{-i\frac{\pi}{3}q_L j} |Q, j\rangle . \tag{22}$$

Eq. (22) indicates that a specific $|q_L\rangle$ state is an equal superposition of the $|j\rangle$ states, analogous to the one studied in Sec. 2.2. Through a permutation $U = U^\dagger$ of the charge ordering, $F^{(6)}$ can be decoupled as $F^{(6)} = U \left( F^{(2)} \otimes F^{(3)\dagger} \right) U^\dagger$ where

$$F^{(2)} = \frac{1}{\sqrt{2}} \begin{pmatrix} 1 & 1 \\ 1 & -1 \end{pmatrix}, \quad F^{(3)\dagger}_{\tilde{q}_L, \tilde{j}} = \frac{e^{i\frac{2\pi}{3}\tilde{q}_L\tilde{j}}}{\sqrt{3}} . \tag{23}$$

Here $F^{(2)}$ is recognised as the Majorana F-matrix [48] and relates $p_L$ and $p_J$ (the parity degrees of freedom). $F^{(3)\dagger}$ is a $\mathbb{Z}_3$ PF $F$-matrix that relates the fractional degrees of freedom $\tilde{q}_L$ and $\tilde{j}$ (see App. D for details). We introduce the reduced density matrices $\tilde{\rho}_L = \text{Tr}_{p_L}[\rho_L]$ and $\tilde{\rho}_J = \text{Tr}_{p_J}[\rho_J]$ which describe the $\tilde{q}_L$ and $\tilde{j}$ degrees of freedom, respectively; they are obtained by tracing the full density matrices expressed in the eigenbasis of $q_L$ and $j$ over the related

parities. As an effect of the decomposition of $F^{(6)}$ into the parity and fractional sectors, these reduced density matrices are related by

$$\tilde{\rho}_L = F^{(3)}\tilde{\rho}_J F^{(3)\dagger}. \tag{24}$$

In App. D we derive this relation explicitly.

A current readout allows us, in the different limits discussed above, to measure the fractional charges $\tilde{q}_L$ and $\tilde{j}$. Therefore, Eq. (24) implies that, in a series of consecutive fusions, the probability of measuring the outcome $\tilde{j}$, conditioned to the fact that the previous measurement returned $\tilde{q}_L$, is given by $|F^{(3)}_{\tilde{j},\tilde{q}_L}|^2$. However, the square amplitude of all elements in $F^{(3)}$ is $1/3$, hence any of the $|\tilde{q}_L\rangle$ states will lead to an outcome of $\tilde{j}$ that is 0, 1 or 2 with the same probability $1/3$. The same holds true the other way around. Similar results were proposed for Majorana modes in Ref. [51]. The equal probability of all outcomes in these alternating measurements is a strong indication of the non-abelian character of PFs, because it derives from the non-trivial commutation rules between the observables $P_L$ and $P_J$. Such a probability distribution would be different from measurement outcomes related to most non-topological degrees of freedom localised in proximity of the lead. For instance, if the PFs in our system were replaced by a single antidot with a localised fractional charge, the current readout would always return the same outcome (up to random poisoning events).

The current readout schemes presented in the previous sections can be adopted to experimentally observe the probabilities $|F^{(3)}_{\tilde{j},\tilde{q}_J}|^2$ in a way reminiscent of measurement-based topological quantum computation [52]. To this end, we propose a quantum quench protocol to first initialise the four-PF system in a state with a well-defined $\tilde{q}_L$ and then measure $\tilde{j}$, thereby obtaining statistics of the amplitude of the reduced associativity matrix $F^{(3)\dagger}$. The protocol requires the ratios $\varepsilon_L/J$ and $\varepsilon_R/J$ to be tunable in order to apply a suitable tunnel spectroscopy readout of $\tilde{j}$. In the next subsection we present a strategy to vary the couplings $\varepsilon_i$ over several orders of magnitude.

We consider the device in Fig. 8, such that a normal metallic lead is connected to $\alpha_2$ with a coupling rate $\Gamma$, as described in the previous sections. We assume that the fractional Josephson coupling $J$ and the rate $\Gamma$ cannot be controlled and are set to values such that $\Gamma \ll J$. The initialisation and readout protocol consist of the following steps:

**At times $t < 0$:** We initialise the setup in the regime $\Gamma \ll J \ll \varepsilon_L^{\text{ini}}, \varepsilon_R^{\text{ini}}$. As described before, the basis $\{|Q, q_L\rangle\}$ approximates the eigenstates of the system and the groundstate carries a major weight in one of the $q_L$ states. To validate the initialisation, the $\tilde{q}_L$ state can be measured through the current across the lead, as discussed in the previous sections. This measurement, however, is not strictly necessary with a strong $\varepsilon_L^{\text{ini}}$ and $\varepsilon_R^{\text{ini}}$, and we assume that the system is in its groundstate.

**At time $t = 0$:** We quench $\varepsilon_L^{\text{quench}}, \varepsilon_R^{\text{quench}} \ll \Gamma \ll J$. The PFs $\alpha_1, \alpha_4$ are almost decoupled from the rest of the system, and the eigenstates of the Hamiltonian are now approximately $\{|Q, j\rangle\}$.

**At $t > 0$:** $\tilde{j}$ is conserved (up to errors of a few percent) on a time scale $\sim 2\pi |\varepsilon_i^{\text{quench}}|^{-1}$ by the same mechanism analysed in Sec. 4, and we read out the corresponding current signal. On average, we expect to measure $\tilde{j} = 0, 1$ and 2 with equal probability $1/3$.

To evaluate this protocol, we use again the quantum jump method and stochastically evolve the system within a sector with well-defined $Q \mod 3$. We initialise the evolution in the groundstate of the Hamiltonian $H_{4\text{pf}}(t < 0)$ and adopt the jump operators defined in Eq. (21) using the eigenstates of $H_{4\text{pf}}(t > 0)$. With the same procedure and time interval ($t_{\text{avg}} = 0.1\,\mu s$) as for the two-PF scenarios, we calculate the current evolution at a fixed voltage bias of the lead. Specifically, for $H_{4\text{pf}}(t < 0)$ we use $\varepsilon_L^{\text{ini}} = 50\lambda$ and $\varepsilon_R^{\text{ini}} = 45\lambda$ whereas the values after the quench are $\varepsilon_i^{\text{quench}} = 10^{-4}\varepsilon_i^{\text{ini}}$. The other parameters are set to:

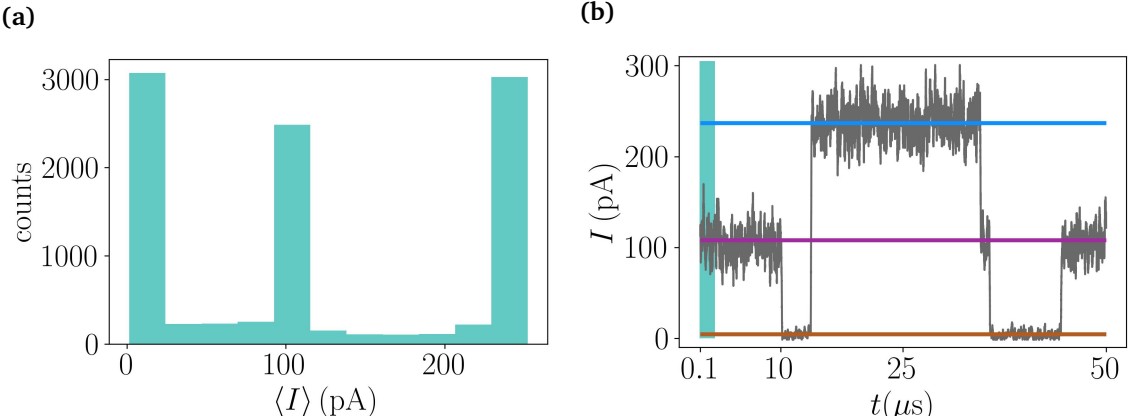

Figure 9: **(a)** Statistics of the mean current $\langle I \rangle$ between $0.1\,\mu s$ and $2\,\mu s$ for $10^4$ simulations of the quench protocol with a binning size of 22.8 pA. The mean current is mainly distributed around the three values of $\langle I \rangle_{\tilde{j}}$, but not evenly due to the finite $J$ at $t < 0$. Counts with $\langle I \rangle$ well between the three "pillars" reflect jumps between $|\tilde{j}\rangle$ states taking place on a $\mu s$ scale. **(b)** Current trajectory on a long time scale compared to $2\pi|\varepsilon_i^{\text{quench}}|^{-1} \approx 0.1\,\mu s$. The current displays jumps between three levels just like in Sec. 3, where the levels correspond to $\langle I \rangle_{\tilde{j}=0}$ (brown line), $\langle I \rangle_{\tilde{j}=1}$ (light blue line) and $\langle I \rangle_{\tilde{j}=2}$ (violet line). The shaded turquoise area indicates the time interval $t_m$, used to calculate $\langle I \rangle$ in **(a)**.

$J = \lambda, \varepsilon_{LR} = 5 \cdot 10^{-4}\lambda, \Gamma = 0.1\lambda, \phi_L = \arctan(1/\sqrt{27}) \approx \pi/16, \phi_R = \pi/14$ and $\theta = \pi/15$. The voltage bias is fixed to $eV_b = 2.6\lambda$ similarly to Secs. 3 and 4. In Fig. 9(a) we show a histogram of the mean value of the current $\langle I \rangle$, numerically observed in the interval $t_m$ from $t = 0.1\,\mu s$ to $t = 2\,\mu s$ ($t_m = 456 \cdot 2\pi\Gamma^{-1} \approx 23 \cdot 2\pi|\varepsilon_i^{\text{quench}}|^{-1}$) for $10^4$ simulations of the quench protocol. The results are seen to bunch into three different values around 0 pA, 100 pA and 250 pA. These values approximately correspond to the mean current of the system with $\varepsilon_i^{\text{quench}} = 0$ when initialised in the eigenstates of $\tilde{j}$; $\langle I \rangle_{\tilde{j}=0} = 4.5$ pA, $\langle I \rangle_{\tilde{j}=1} = 237$ pA and $\langle I \rangle_{\tilde{j}=2} = 108$ pA (calculated by averaging over $50\,\mu s$). In an ideal case where the fractional Josephson coupling is turned off completely at times $t < 0$ and only acquires a finite value at $t = 0$, and where $\varepsilon_i^{\text{quench}} = 0$, the distribution among the sectors $\tilde{j} = 0, 1, 2$ is indeed equal with $1/3$ probability for each. In the more realistic scenario simulated here, we observe two different deviations from the ideal result. Firstly, the finite $J$ at $t < 0$ implies that the initial groundstate is not a perfectly defined $|q_L\rangle$ eigenstate, and this is the reason for the uneven distribution among sectors despite the large number of simulations. Secondly, a finite background of counts appears at intermediate $\langle I \rangle$ between the three values of $\langle I \rangle_{\tilde{j}}$. This reflects that jumps between different $|\tilde{j}\rangle$ states take place in some of the trajectories within the $t_m = 1.9\,\mu s$ that the current is averaged over, due to the finite $\varepsilon_i^{\text{quench}}$. Indeed, $t_m$ is considerably larger than the scale $2\pi|\varepsilon_i^{\text{quench}}|^{-1}$, and the quasiparticle poisoning determined by the coherent $\varepsilon_i^{\text{quench}}$ perturbations is not negligible (we stress that $t_m \neq t_{\text{avg}} = 0.1\,\mu s$). Despite this, the distribution in Fig. 9(a) displays three clearly separated peaks with only a weak background of counts.

In Fig. 9(b) we show an example of the current trajectory over long times compared to the time interval $t_m$ (indicated by the shaded area). Indeed we see a three-level telegraph noise just as in the 2PF case of Sec. 3, which typically occurs on time scales longer than $t_m$. Finally, we note that based on our findings in Sec. 2.2 the projection into one of the $\tilde{j}$ sectors happens on a time scale much shorter than $0.1\,\mu s$, such that the current readout provides clear signatures of the $\tilde{j}$ sectors.

## 5.2 Physical construction of four-parafermion device

Inspired by Refs. [18, 44–46, 53], we propose in this section a construction of a four-PF system with tunable nearest neighbour PF coupling, based on transmon technology. Referring to Fig. 8, each of the two SC islands display a charging energy $E_{C,i}$, $i = L, R$ as they are not directly connected to ground. In addition, they have also a mutual charging energy $E_{CC}$ determined by a cross capacitance. The two SCs are connected through a fractional Josephson coupling $J$ mediated by the PFs $\alpha_2$ and $\alpha_3$ [7, 46, 47]. Both islands are also coupled through an integer Josephson coupling $E_{J,i}$ to a background grounded SC. We assume that the Josephson couplings $E_{J,i}$ can be tuned, for example by using a gate-controlled nanowire junction (gatemon) [54–56]. The related Hamiltonian reads

$$
\begin{aligned}
H_{\text{trans}} = \sum_{i=L,R} & \Big[ E_{C,i}\big(N_i + q_i/3 + q_{\text{ind},i}\big)^2 - E_{J,i}\cos(\vartheta_i) \Big] \\
& - J\Big( e^{-i\left(\frac{\pi}{6}+\theta\right)} e^{i\vartheta_R \delta_{q_R,0}} \alpha_3^\dagger \alpha_2 \, e^{-i\vartheta_L \delta_{q_L,0}} + \text{H.c.} \Big) \\
& + E_{CC}\big(N_L + q_L/3 + q_{\text{ind},L}\big)\big(N_R + q_R/3 + q_{\text{ind},R}\big) \, .
\end{aligned}
\tag{25}
$$

Here $N_i = -2i\partial_{\vartheta_i}$ is the charge of the Cooper pairs in the SC island $i$ and $\vartheta_i$ is the corresponding phase operator, such that $e^{-i\vartheta_i}$ is the annihilation operator for a Cooper pair in the SC $i$. $q_{\text{ind},i}$ is the tunable charge induced on the SC island $i$ by an external voltage gate, $eq_i/3$ is the charge of the PF pair in island $i$. The phase $\theta = \pi\Phi/3\Phi_0$ is proportional to the magnetic flux $\Phi$ threading the superconducting circuit composed by the two SC islands and the background SC in Fig. 8. It can thus be tuned by small variations of the magnetic field which do not affect the state of the FQH system. $\Phi_0$ is the magnetic flux quantum. The fractional Josephson junction term proportional to $J$ includes operators $e^{\pm i\vartheta_i \delta_{q_i,0}}$ that account for the creation and annihilation of a Cooper pair resulting from the tunnelling of a charge $e/3$ when the PFs in island $i$ share, respectively, the state $q_i = 5$ or $q_i = 0$ (see Refs. [57, 58] for analogous constructions with Majorana modes).

By generalising the Majorana calculation in Refs. [53, 58], we obtain that, in the transmon regime $E_{J,i} \gg E_{C,i}$, the low-energy dynamics of the system can be approximated by the semiclassical effective Hamiltonian in Eq. (16) with the following parameters:

$$
\varepsilon_i \propto E_{C,i}^{1/4} E_{J,i}^{3/4} \, e^{-\sqrt{8E_{J,i}/E_{C,i}}} \, ,
\tag{26}
$$

$$
\varepsilon_{LR} \propto \tilde{E}_C^{1/4} \tilde{E}_J^{3/4} \, e^{-\sqrt{8\tilde{E}_J/\tilde{E}_C}} \, ,
\tag{27}
$$

$$
\phi_i = \pi q_{\text{ind},i} \, ,
\tag{28}
$$

with $\tilde{E}_J \approx 2E_J\cos(\theta)$ and $\tilde{E}_C \approx (E_{CC} + 2E_C)/4$ in the limit $E_{J,L} = E_{J,R} = E_J$ and $E_{C,L} = E_{C,R} = E_C$. $\varepsilon_i$ is calculated based on the $2\pi$ phase slip of island $i$ and neglecting the $E_{CC}$ pairing. More refined results can be found in Ref. [53]. The term $\varepsilon_{LR}$, instead, accounts for the simultaneous phase slip by $2\pi$ of both the islands estimated by imposing the freezing of the phase difference $\vartheta_L - \vartheta_R = 0$, and it becomes sizeable for devices with large $E_{CC}$. The phases $\phi_i$ in Eqs. (16) and (18) are proportional to the charges induced on the two SC islands and can in principle be controlled through suitable voltage gates. All the previous estimates have been derived by neglecting the fractional Josephson term $J \ll E_J$. Importantly, this semiclassical analysis shows that the couplings $\varepsilon_i$ in Eq. (26) can be exponentially varied by changing the ratio $E_{J,i}/E_{C,i}$, as common in transmon devices.

o estimate realistic bounds for the variation of the couplings $\varepsilon_i$, we consider the Josephson energies reported in Ref. [54] which can reach 1 THz, and we use the charging energies for aluminium islands of similar size as those in [1,2], around 0.1 meV $\approx$ 25 GHz [59]. This yields a ratio in the exponent of Eq. (26) around $E_J/E_C \approx 40$. We consider a minimum value $E_J = 0.1$

THz (below which the semiclassical approximation fails). In this case, by using voltage gates to pinch off or open the Josephson junctions, we obtain a ratio between minimum and maximum values of the PF coupling given by $\varepsilon_i(E_J = 0.1 \text{ THz})/\varepsilon_i(E_J = 1 \text{ THz}) \approx 2.7 \cdot 10^{-5}$, which justifies the more conservative ratio $10^{-4}$ adopted to derive the data in Fig. 9(a).

Alternative four-PF devices with tunable couplings can be used for similar quench and readout protocols such as for instance PFs embedded in fluxonium circuits [47]. Furthermore, systems with a tunable coupling rate $\Gamma$ between the lead and the PFs could be exploited to perform more efficient measurement protocols for the associativity rules and to extend the validity of our analysis to a broader range of parameters.

## 6 Discussion and conclusions

We have analysed the transport signatures of $\mathbb{Z}_6$ PFs in hybrid FQH-SC devices coupled to a normal electrode and demonstrated that current readouts can reveal the fractional character of these modes in suitable ranges of parameters. Differently from previous works focusing on the transport of fractional charges [43, 46, 60–64], all the measurements proposed here rely on the tunnelling spectroscopy of PF systems mediated by electrons. Our analysis is based on a quantum jump approach to simulate single stochastic trajectories; it shows that current signals can distinguish the different values of the fractional charge shared by two PFs under realistic conditions, thus providing a readout method alternative to interferometric techniques [17,43]. The quantum jump simulations allowed us to derive how the current measurements become projective measurements of the parafermionic degree of freedom, to estimate the time required for this readout and to investigate the related noise.

We extended the jump operator formalism to describe also the incoherent coupling with a bath of fractional charges. In particular, this enabled us to model the fractional quasiparticle poisoning caused by the interaction of the PFs with the external edge modes of the FQH system, modelled as a chiral Luttinger liquid. As a result, we confirmed the onset of an expected three-level telegraph noise [11] of the current for weak poisoning rates reflecting the jumps of fractional charges to and from the PF system over time. A telegraph current noise signal could also be observed when coherently coupling the pair of PFs to an antidot. For small antidots the result is a two-level noise signal. Inspired by Ref. [43], we remark that a deliberate construction of an antidot in the FQH platform (as demonstrated in Ref. [12]) can be used to extend the results and protocols presented in Secs. 4 and 5 to detect further signatures of PFs and manipulate them. For example, devices with tunable antidot energy levels ($\delta_{\text{ad}}$ in our model) could be used to dynamically control the transitions between the charge states of the PFs; the related manipulation protocols would be similar to the ones proposed to observe a topological blockade effect in non-abelian FQH states [65], or to evolve the states of Majorana modes through their coupling with quantum dots [66–68]. Moreover, additional transport signatures of the presence of PFs based on the devices presented in Sec. 4 can be offered by the comparison of antidots of different sizes: the two-level telegraph noise behaviour of the current for weak PF-antidot coupling would evolve into the three-level noise when enlarging the antidot, thus lowering its energy splittings and involving more charge states in the system dynamics.

To study the anyonic properties of parafermionic modes, we expanded the analysis to devices with four PFs. We proposed a quench protocol to test the associativity rules of the fusion of $\mathbb{Z}_6$ PFs and, using again the quantum jump method, we obtained statistics on the amplitude of the $F$-matrix corresponding to their fractional $\mathbb{Z}_3$ degree of freedom. Our protocol is based on the possibility of tuning some of the couplings between the PFs, and we presented a physical device based on transmon-like elements to achieve the required control over these

interactions.

In general, our simulations are suitable to study also more general noise effects, relevant for experimental implications, including the consequences of drifting parameters such as the magnetic flux.

Our results provide indications for the design of hybrid FQH-SC platforms aimed at detecting PFs and studying some of their anyonic features. Furthermore, the readout schemes we analysed can be integrated in more complex setups to implement their braiding [16,17] thus offering additional tools for their study.

## Acknowledgements

We thank A. C. C. Drachmann, K. Flensberg, A. Maiani, C. Marcus, M. Mintchev, F. Nathan and M. M. Wauters for helpful discussions, insightful observations and technical advice.

**Funding information** I. E. N. and M. B. are supported by the Villum foundation (research Grant No. 25310). We acknowledge support from the Deutsche Forschungsgemeinschaft (DFG) project Grant No. 277101999 within the CRC network TR 183 (subproject C01), as well as Germany's Excellence Strategy Cluster of Excellence Matter and Light for Quantum Computing (ML4Q) EXC 2004/1 390534769 and Normalverfahren Projektnummer EG 96-13/1. This project also received funding from the European Union's H2020 research and innovation program under grant agreement No. 862683. J. S. also acknowledges funding from the Danish National Research Foundation, the Danish Council for Independent Research | Natural Sciences and from the Knut and Alice Wallenberg Foundation through the Academy Fellow program.

## A   The quantum jump method

Here we describe the quantum jump method, including the Monte Carlo technique, used to calculate trajectories and the quantities derived from these.

The total time of the simulation is divided into small time steps $\delta t$ such that $\delta t \ll 2\pi\Gamma^{-1}$. Within each interval the probability that an electron enters or leaves the PF system through the metallic lead is thus much smaller than 1. As stated in the main text, the coupling rate is taken to be $\Gamma = 2\pi \cdot 0.24$ GHz and we therefore use $\delta t = 0.1$ ns. With the Hamiltonian $H_{2\text{pf}}$ in Eq. (1) and the jump operators in Eq. (2) we define the no-jump evolution operator

$$\mathcal{U} = \exp\left[-i\delta t\left(H_{\text{pf}} - \frac{i}{2}\sum_{s=\pm}\left(L_s^e\right)^\dagger L_s^e\right)\right].\tag{29}$$

We need this to calculate the trajectories for which the protocol is:

**Step 0:** The state is initialised in a superposition of charge eigenstates $|\psi(t=0)\rangle = \sum_q c_q |q\rangle$ with coefficients $c_q$ that may all be non-zero.

**Step 1:** The no-jump evolution operator in Eq. (29) is applied to the state $|\psi(t)\rangle$ with subsequent renormalisation due to the non-unitarity of $\mathcal{U}$. The renormalised state is labelled $|\psi(t+\delta t)\rangle$.

**Step 2:** We draw two random numbers from a uniform distribution, $p_{s=\pm} \in [0,1]$, and we furthermore randomly order $L_+^e$ and $L_-^e$.

**Step 3:** We calculate the jump probabilities $P_s(t+\delta t) = \langle\psi(t+\delta t)|(L_s^e)^\dagger L_s^e|\psi(t+\delta t)\rangle \delta t$. By the order set in step 2, we first check if $p_{s'} < P_{s'}(t+\delta t)$ where $s'$ is the sign set by the random ordering. If this is the case, we apply the corresponding jump operator $L_{s'}^e$ to the

state $|\psi(t+\delta t)\rangle$ and renormalise. If not, we check the probability of a jump in the opposite direction $(-s')$ and if $p_{-s'} < P_{-s'}(t+\delta t)$ we apply the jump operator $L^e_{-s'}$ with subsequent renormalisation. In case neither of the conditions are fulfilled, we keep the state $|\psi(t+\delta t)\rangle$. The protocol is repeated from step 1 with the state obtained in step 3 until the final simulation time is reached. The assumption of Markovianity of the electron bath means that the distribution function of the lead is not changed by a jump, but remains the equilibrium Fermi-Dirac function.

We note that this approach is distinct from the stochastic Schrödinger equation [15] in the way that the randomly applied jump operators yield an immediate, discontinuous state transition from $|\psi(t+\delta t)\rangle$ to $L^e_\pm|\psi(t+\delta t)\rangle$ whereas in the stochastic Schrödinger equation, the jump operators act as weak projections yielding a continuous state transition.

At every time step of the trajectories we can calculate the expectation value of the charge $\langle\psi(t)|q|\psi(t)\rangle$ as well as the participation ratio $\zeta$ introduced in Eq. (4). Furthermore, by keeping count of the times that the jump operators are applied, we can also estimate the current. We introduce the number $j_{\text{diff}}(t)$ which is 0 at $t=0$ and increases by one whenever $L^e_+$ is applied to the state and decreases by one if $L^e_-$ is applied. The current in the lead is simply obtained by $I(t) = |e|(j_{\text{diff}}(t) - j_{\text{diff}}(t - t_{\text{avg}}))/t_{\text{avg}}$ where $t_{\text{avg}}$ is an integration time which we set to $t_{\text{avg}} = 0.1\,\mu s$. This is how Fig. 3(a) has been obtained.

In the system analysed in Sec. 3, the coupling with the edge modes is introduced through the use of the two additional jump operators $L^{e/3}_\pm$ in Eq. (10). In this case, we apply the same approach described by the steps 0-3, with a randomised ordering of all four jump operators in steps 2 and 3.

To give a rough estimate of the $\tilde{q}$ projection time discussed in Sec. 2.2, we analyse the continuous part of the wave function evolution and consider in particular the part of $\mathcal{U}$ that contains the product of jump operators $L^e_\pm$. This term in the exponent is diagonal with elements

$$-\delta t\,\Gamma\left(J_+(\Xi_{q+3}, V_b) + J_-(\Xi_{q+3}, V_b)\right)/2 \equiv -\delta t\,\tilde{\lambda}(q) \tag{30}$$

at the $(q, q)$ entry and thus leads to a decay which partly compensates for the fact that the jump operators themselves, when applied in step 3, project with different weights into the different states $|q\rangle$. The operator $\mathcal{U}$ allows us to determine the projection time for the special trajectory in which no jumps occur by comparing the decay rates $\tilde{\lambda}(q)$ of all the charge eigenstates. The two states with the lowest $\tilde{\lambda}(q)$ are $|q=0\rangle$ and $|q=5\rangle$; hence, in the absence of jumps, the initial state $|\text{equal}\rangle$ can be considered projected when

$$\left(\frac{e^{-\tilde{\lambda}(5)\delta t}}{e^{-\tilde{\lambda}(0)\delta t}}\right)^N \ll 1, \tag{31}$$

where $N$ is the number of time steps such that $\delta t \cdot N$ is the projection time and $0 < \tilde{\lambda}(0) < \tilde{\lambda}(5)$. Eq. (31) provides a qualitative indication of the behaviour of the projection time as a function of the voltage bias and temperature which, as expected, is proportional to $\Gamma^{-1}$. In Fig. 10(a) we compare the trend for a no-jump evolution with the data obtained for stochastic trajectories by superimposing the function $(\tilde{\lambda}(5) - \tilde{\lambda}(0))^{-1}$, scaled by a numerical factor $1.45 \cdot 10^{-3}$, on the data of Fig. 2(b). Indeed there is a qualitative agreement between this estimate and the inferred projection time from the $\langle\tilde{q}\rangle$ and $\zeta$ measures at voltages $|eV_b| \lesssim |\Xi_q|$. The discrepancy at larger energies is due to fact that this estimate disregards the actual quantum jumps which dephase the state. As mentioned in the main text the state $|\text{equal}\rangle$ is a worst case example for the projection time since it includes all six charge eigenstates which have competing decay rates. The projection of the simpler initial state $\frac{1}{\sqrt{3}}(|q=0\rangle + |q=1\rangle + |q=2\rangle)$ into one of the charge sectors is faster which we see already on the short $O(2\pi\Gamma^{-1})$ time scale (compare Fig. 10(b) with Fig. 2(a)).

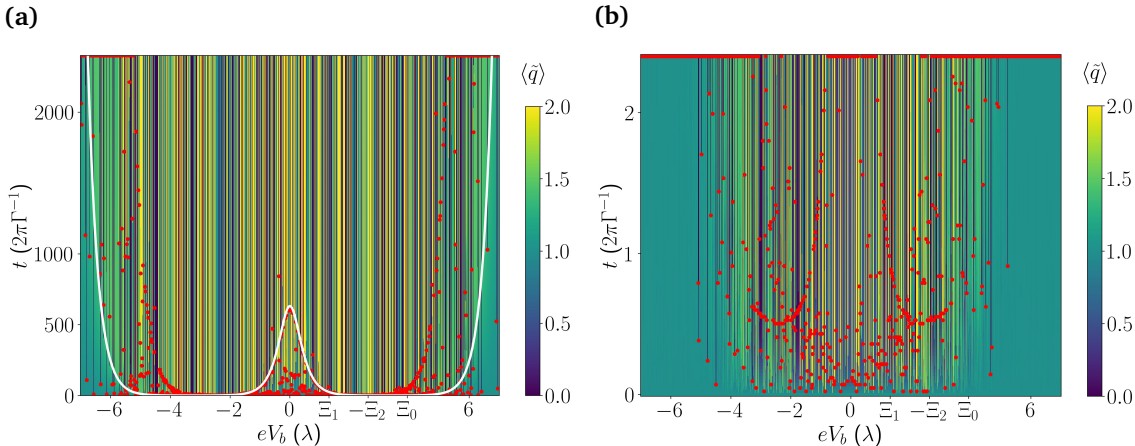

Figure 10: **(a)** The plot in Fig. 2(b) with the white curve $1.45 \cdot 10^{-3} \left( \tilde{\lambda}(5) - \tilde{\lambda}(0) \right)^{-1}$ superimposed. **(b)** Similarly to Fig. 2(a), this is the evolution of $\langle \tilde{q} \rangle$, but with the simpler initial state $\frac{1}{\sqrt{3}} (|q = 0\rangle + |q = 1\rangle + |q = 2\rangle)$. The meaning of the red dots is as in the main text and all parameters are the same.

# B   Fractional jump operators and anyonic distribution function

We derived the modelling of fractional quasiparticle poisoning in Sec. 3 from the effective tunnelling of fractional charges between the quantum Hall edge and the PFs given by $H_{c,\text{edge}}$ in Eq. (8). To model this coupling, we introduced the FQH edge operator:

$$\psi_{e/3} = \left( \frac{\Delta_{\text{FQH}}}{2\pi} \right)^{\frac{1}{6}} e^{i \varphi_r(x=0)} . \tag{32}$$

The operator $\psi_{e/3}$ annihilates a charge $e/3$ at an arbitrary position $x = 0$ of the FQH edge, and it is expressed as the vertex operator of a suitable chiral bosonic field $\varphi_r(x)$ [69]. Assuming that the coupling with the PFs is sufficiently weak, such that it does not affect the edge mode properties, the $\psi_{e/3}$ two-point correlation function in time for inverse temperature $\beta$ and chemical potential $\mu$ results

$$\left\langle \psi_{e/3}^{\dagger}(t) \psi_{e/3}(0) \right\rangle = \frac{e^{-it\mu}}{\left( 2i \Delta_{\text{FQH}} \beta \sinh \frac{\pi t}{\beta} \right)^{1/3}} . \tag{33}$$

Experimental observations of the heat flow of FQH edge modes in GaAs suggest considerably fast relaxation times for the $\nu = 1/3$ state [70] compared to the typical scale $2\pi \Gamma_{e/3}$ in our simulations. We assume an analogous behavior for the FQH edge modes in graphene, such that we treat them as a Markovian bath. By following Ref. [26] we can thereby define jump operators originating from $H_{c,\text{edge}}$ as

$$L_{\pm}^{e/3} = \sum_{n,m} \sqrt{2\pi |\eta_{e/3}|^2 d(E_n - E_m, \pm \mu)} \, \langle m | \alpha_2^{\mp 1} | n \rangle \, | m \rangle \langle n | , \tag{34}$$

where $|n\rangle$ and $|m\rangle$ label the eigenstates of the two-PF Hamiltonian $H_{\text{pf}}$ (Eq. (1)), which are also eigenstates of the charge $q$, and $\alpha_i^{-1} = \alpha_i^{\dagger}$. In Eq. (34) the function $d$ is the spectral function associated with the operator $\psi_{e/3}$:

$$d(E, \mu) = \int_{-\infty}^{+\infty} dt \, e^{iEt} \left\langle \psi_{e/3}^{\dagger}(t) \psi_{e/3}(0) \right\rangle . \tag{35}$$

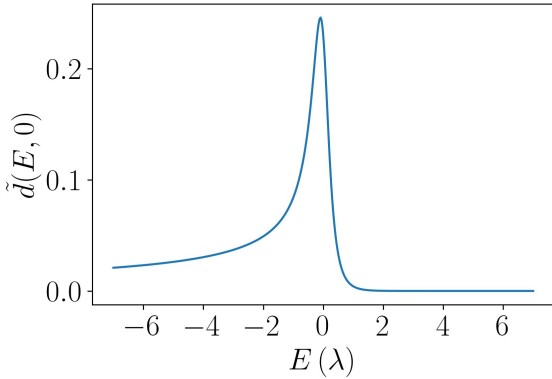

Figure 11: The anyonic spectral function $\tilde{d}(E, \mu)$ for $\mu = 0$ at a temperature $T = \lambda/3k_{\mathrm{B}} \approx 40$ mK ($k_{\mathrm{B}} \neq 1$).

The spectral function for $\psi_{e/3}^{\dagger}$ is simply $d(E, -\mu)$. From the results by Liguori, Mintchev and Pilo [38] and by redefining $d(E, \mu) = \beta \tilde{d}(E, \mu)$ we derive Eq. (9). The explicit expression for Eq. (34) depends on the specific choice of the PF charge eigenstates $|q\rangle$. We adopt the convention $|q\rangle = (\alpha_1^{\dagger})^q |q = 0\rangle$. In this case, the inner products appearing in Eq. (34) read

$$\langle q|\alpha_2|q'\rangle = \mathrm{e}^{i\pi/6}\,\mathrm{e}^{i\pi(q'-1)/3}\,\delta_{q,q'-1}\,, \tag{36}$$

$$\langle q|\alpha_2^{\dagger}|q'\rangle = \mathrm{e}^{-i\pi/6}\,\mathrm{e}^{-i\pi q'/3}\,\delta_{q,q'+1}\,, \tag{37}$$

and, replacing these expressions in Eq. (34), we obtain Eq. (10). The Eqs. (36) and (37) are also used to obtain the matrix elements in Eq. (14).

The anyonic distribution function $d(E, \mu)$ for particles with charge $e/3$ displays, at finite temperature, the typical behaviour depicted in Fig. 11. It is characterized by a peak in $E = \mu$, which diverges as $\Delta_{\mathrm{FQH}}^{-1/3}(\mu - E)^{-2/3}\Theta(\mu - E)$ in the zero-temperature limit $\beta \to \infty$. This divergence is analogous to the divergence of the Bose-Einstein distribution for bosonic condensates. Eq. (10) allows us to examine the impact of this divergence on the quasiparticle poisoning rate. In the zero-temperature limit, the poisoning rate is weak compared to the inverse of the projection time $t_{\mathrm{proj}}^{-1}$ if

$$\Delta_{\mathrm{FQH}}^{-1/3}\delta\mu^{-2/3}|\eta_{e/3}|^2 \ll t_{\mathrm{proj}}^{-1} \approx \frac{\Gamma}{40\pi}\,, \qquad \text{for } T = 0\,, \tag{38}$$

where $\delta\mu$ is the minimum of the displacements $E_{q\mp1} - E_q \mp \mu$ from the chemical potential, and the projection time is estimated from the data in Fig. 2(b) in the favourable range of voltages around the energies $\pm\Xi_q$. Since $\mu$ can be varied by changing the voltage of the system edge, Eq. (38) indicates that the zero-temperature resonances can be avoided by tuning the system to sufficiently large $\delta\mu$. Specifically, the data presented in Fig. 5 are calculated by considering a typical FQH gap of graphene $\Delta_{\mathrm{FQH}} = 1.7$ meV [41], $\Gamma = \lambda/10$ and $\eta_{e/3} = 3.5 \cdot 10^{-3}\lambda$ with $\lambda = 2\pi \cdot 2.4$ GHz which result in the constraint $\delta\mu \gg 2 \cdot 10^{-3}\lambda^{3/2}/\Delta_{\mathrm{FQH}}^{1/2} \approx 1.5 \cdot 10^{-4}\lambda \approx 1.4 \cdot 10^{-6}$ meV.

At finite temperatures, however, the resonances simply correspond to peaks of $d(E, \mu)$ as in Fig. 11 and their effect is thus limited. In particular, by considering an explicit resonant value of $d$ at $E = \mu$, we obtain the following condition for the weak quasiparticle poisoning regime:

$$\frac{\Gamma^2\left(\frac{1}{6}\right)|\eta_{e/3}|^2\beta^{2/3}}{2\pi\Gamma(\frac{1}{3})\Delta_{\mathrm{FQH}}^{1/3}} \approx 1.84\frac{|\eta_{e/3}|^2\beta^{2/3}}{\Delta_{\mathrm{FQH}}^{1/3}} \ll t_{\mathrm{proj}}^{-1} \approx \frac{\Gamma}{40\pi}\,, \qquad \text{at } \delta\mu = 0\,. \tag{39}$$

**(a)**                                                **(b)**

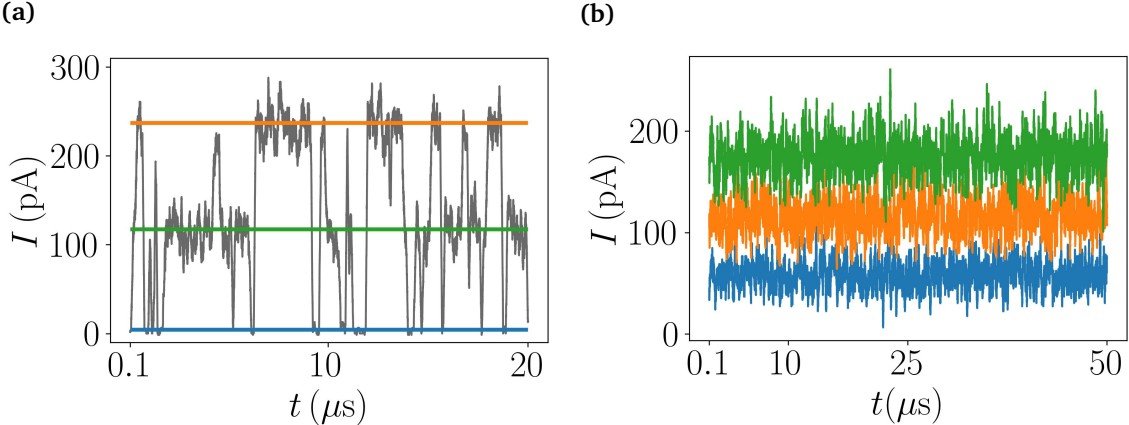

Figure 12: **(a)** Current evolution for two PFs coupled to the external edge modes of the FQH liquid with a strength $\eta_{e/3} = 10.5 \cdot 10^{-3} \lambda$. All other parameters are the same as in Fig. 5. **(b)** Current evolution of the setup described in Sec. 4 with a PF-antidot coupling $\eta_{\text{ad},i} = 10\Gamma$. The three colours correspond to the three initial charge sectors $\tilde{q} = 0$ (blue), $\tilde{q} = 1$ (orange) and $\tilde{q} = 2$ (green).

For the previous choice of the physical parameters, this corresponds to $T \gg 0.04$ mK which is fulfilled by more than two orders of magnitude in realistic experimental conditions ($T \gtrsim 10$ mK).

## C   Strong environment coupling

This appendix provides additional figures presenting current signatures where the coupling between the PFs and the environment is stronger than in the figures of the main text, both for the incoherent coupling with edge modes and for the coherent coupling with an antidot.

In Fig. 12(a) we show the current trajectory for the system described in Sec. 3 with the same parameters as in the main text except from the coupling strength to the fractional edge modes $\eta_{e/3}$ which is increased by a factor of 3 such that $\Gamma_{e/3} \approx 0.02\Gamma$. At this coupling rate, the $\tilde{q} = 0$ sector is occupied for periods so short, compared to the integration time, that it becomes problematic to distinguish this current signal from noise.

In Fig. 12(b) current trajectories are shown for the three initial states $\frac{1}{\sqrt{2}}(|0,0\rangle + |3,0\rangle)$ (blue), $\frac{1}{\sqrt{2}}(|1,0\rangle + |4,0\rangle)$ (orange) and $\frac{1}{\sqrt{2}}(|2,0\rangle + |5,0\rangle)$ (green) where the coupling strengths between the PFs and the antidot are $\eta_{\text{ad},1} = \eta_{\text{ad},2} = 10\Gamma$. Apart from this, all parameters are the same as in Fig. 7. The three levels of the current correspond to the most dominant transition within each of the sectors with conserved total fractional charge: $(\tilde{q} + q_{\text{ad}}) \mod 3$.

## D   Details on the F-matrix

The permutation $U$ introduced in Sec. 5.1 swaps the states $|q_L = 1\rangle = |p_L = 1, \tilde{q}_L = 1\rangle$ and $|q_L = 4\rangle = |p_L = 0, \tilde{q}_L = 1\rangle$ and likewise in the $|j\rangle$ basis. It reads

$$
U = \begin{pmatrix} 1 & & & & & \\ & 0 & & & 1 & \\ & & 1 & & & \\ & & & 1 & & \\ & 1 & & & 0 & \\ & & & & & 1 \end{pmatrix}. \tag{40}
$$

Thus, it reorders from the basis $\{|q_L\rangle\} = \{|0\rangle, |1\rangle, |2\rangle, |3\rangle, |4\rangle, |5\rangle\}$ to the ordering $\{|p_L, \tilde{q}_L\rangle\} = \{|0,0\rangle, |0,1\rangle |0,2\rangle, |1,0\rangle, |1,1\rangle, |1,2\rangle\}$ and likewise for the $|j\rangle$ and $|p_J, \tilde{j}\rangle$ states.

Since the current measurements detect only the fractional part of the PF state, we need to trace out the fermionic degree of freedom, $p_J$ or $p_L$, to obtain a $3 \times 3$ reduced density matrix as a function of the fractional charge $\tilde{j}$ or $\tilde{q}_L$ only. In the following, we show that is it possible to define a reduced $3 \times 3$ matrix $F_{\text{red}}$ such that $\tilde{\rho}_L = F_{\text{red}} \tilde{\rho}_J F_{\text{red}}^\dagger$ and that $F_{\text{red}} = F^{(3)\dagger}$, as assumed in the main text. Based on the definition of the matrix $F^{(6)}$ in Eq. (22), this would correspond to

$$
\sum_{p_L = 0,1} \langle p_L | F^{(6)} \rho_J F^{(6)\dagger} | p_L \rangle = F_{\text{red}} \left( \sum_{p_J = 0,1} \langle p_J | \rho_J | p_J \rangle \right) F_{\text{red}}^\dagger. \tag{41}
$$

To demonstrate that this relation indeed matches Eq. (24), we explicitly write out the reduced density matrix related to $\tilde{q}_L$, $\tilde{\rho}_L$, which is obtained by tracing the full density matrix of the four-PF system over $p_L$:

$$
\begin{aligned}
\tilde{\rho}_L = \text{Tr}_{p_L} [\rho_L] &= \text{Tr}_{p_L} \left[ U \left( F^{(2)} \otimes F^{(3)\dagger} \right) U^\dagger \rho_J U \left( F^{(2)\dagger} \otimes F^{(3)} \right) U^\dagger \right] \\
&= F^{(3)\dagger} \left[ \sum_{p_L} \langle p_L | \left( F^{(2)} \otimes \mathbb{1} \right) U^\dagger \rho_J U \left( F^{(2)\dagger} \otimes \mathbb{1} \right) | p_L \rangle \right] F^{(3)} \\
&= F^{(3)\dagger} \left[ \sum_{p_J} \langle p_J | U^\dagger \rho_J U | p_J \rangle \right] F^{(3)} = F^{(3)\dagger} \tilde{\rho}_J F^{(3)}. 
\end{aligned} \tag{42}
$$

In the second line, we have used that the trace over $p_L$ is independent of the cyclic permutation of U (which only interchanges states with the same $\tilde{q}_L$) and that $U^\dagger = U^{-1}$. Furthermore, we adopt the convention that $\rho_J$ and $\rho_L$ are expressed in the $|j\rangle$ and $|q_L\rangle$ basis, respectively, whereas the explicit summation over $p_L$ is done with the states ordered in the $|p_L, \tilde{q}_L\rangle$ basis. $F^{(3)}$ is not affected by the reordering of the $p_L$ degrees of freedom within the $\tilde{q}_L = 1$ sector (by the permutation $U$), and it can be moved out of the partial trace on both sides. In the third line, we have used $F^{(2)}$ to transform from the $|p_L\rangle$ to $|p_J\rangle$ basis and are left with the reduced density matrix $\tilde{\rho}_J$ in the basis $\{|\tilde{j}\rangle\} = \{|0\rangle, |1\rangle, |2\rangle\}$. The above equation confirms that $F^{(3)\dagger}$ is the associativity mapping between the reduced degrees of freedom $\tilde{q}_L$ and $\tilde{j}$.

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
