# Peer review of "Dynamics of parafermionic states in transport measurements"

_SciPost Physics_

## Round 1 · Referee Report · Anonymous (Referee 1) · 2023-6-26

Report

This paper is an extension of previous work (Ref. [11]), which proposes using transport measurements to detect signatures of emergent parafermion zero-modes in hybrid fractional quantum Hall-superconductor structures.

The first portion of the paper serves as a broad examination of effects not previously considered in Ref. [11], mainly considering dynamics out of equilibrium. The authors calculate the evolution of the system using a quantum jump method, calculated numerically via Monte Carlo simulations. Using these techniques the authors examine coherent and incoherent quasiparticle poisoning from multiple sources, such as the external edges of the fractional quantum Hall system, or antidots in the adjacent fractional quantum Hall bulk. The final portion of the paper is dedicated to investigating systems of four parafermion zero-modes, and suggestions of a protocol to study their associativity rules.

The paper is comprehensive and well written, and considers the task at hand from a wide variety of angles. While parafermion zero-modes have yet to be observed experimentally, the ongoing search for simpler Majorana zero modes indicates that there are many sources of uncertainty to consider. Any future search for parafermion zero-modes will thus require both creative experimental signatures, and a thorough theoretical treatment of sources of uncertainty in such experiments. This paper does well to move both accounts forward, and I would thus recommend it for publication.

Granting that, one general comment on the paper, is that it consists of many minute details. In particular, the results are heavily dependent on a large number of parameters, such as the pairing amplitudes ε,η, and η_(e/3), the phase ϕ, the coupling rate Γ, and the energy scale λ. This makes for a somewhat technical read. The authors do well alleviate this effect by writing a well-organized paper, with dedicated separate sections to different effects. They further obtain results for specific values of the parameters described above, which are chosen rather reasonably. It would nonetheless behoove the paper to discuss a bit more broadly the robustness of their results to changes in these parameters, as several of them are dependent on microscopic details and will thus not be tunable experimentally.

A few more specific questions: - In Fig. 5, the authors present jumps between current levels of q ̃=0,1,2. They find that the sector q ̃=0 is less frequently occupied than the other two. Is there a physical intuition behind this result? - The authors use an anyonic spectral function, given in Eq. 9, which affects the results of quasiparticle poisoning through the definition of the jump operators L_±^(e/3). This spectral function corresponds to an anyonic two point function of 〈ψ^† (τ)ψ(0)〉∝τ^(-2δ), with the scaling dimension δ=1/6. Given the difficulty of measuring the scaling dimension, and proposals that it may renormalize via interactions (see, for example, Phys. Rev. Lett. 93, 126801), recent works focusing on the universality of the scaling dimension have asked whether modern experiments typically measure different values of this scaling dimension. How do the results change if the anyonic spectral function changes? - Section 5 consists of many physical limits, and a time-dependent protocol in which the system moves between limits using a quench. I believe that it would serve the paper well to add a second schematic figure, in the vein of Fig. 8, describing the protocol before, during and after the quench. While this won’t add information that isn’t implicitly present in Fig. 8, this will make the protocol easier to understand. - A minor typo: the second paragraph after Eqs. (26)-(28) begins with “o” – this should probably be “To”.

  • validity: -
  • significance: -
  • originality: -
  • clarity: -
  • formatting: -
  • grammar: -

Author:  Ida Nielsen  on 2023-08-23  [id 3919]

(in reply to Report 1 on 2023-06-26)
Category:
answer to question

We thank the referee for the thorough reading and constructive comments. We deeply appreciate the recommendation for publication.

We agree with the referee that this paper and the analysed setups rely on many details and a vast set of parameters in a way that makes immediate simple conclusions hard to extract. To help to the reader obtain an overview of all the considered couplings, interactions and rates, we have included an appendix (now App. A) that provides a list of all the relevant parameters describing their role, realistic range and main effects. In this list, we also briefly discuss which energy scales can be modified in the physical setups and how. Concerning the robustness of the discussed features, the majority of our results does not rely on a precise fine-tuning of the parameters. Their robustness, however, is quite difficult to be quantitatively analysed as the dependence on one parameter may be strongly affected by others. In the manuscript, we have reformulated the conclusion in the end of Sec. 4 to state more clearly the conditions for the robustness of the telegraph noise.

In line with the above, it is difficult to obtain an intuitive explanation of the fact that $\tilde{q}=0$ in Fig. 5 is less occupied. As we discussed, this can be explained based on the steady state obtained by the rate equations. However, this particular sector is a result of the chosen values of $\phi$ and $\mu$ which enter the fractional jump operator in Eq. 10. For differently chosen $\phi$ or $\mu$ it can be one of the other charge sectors showing low occupation. We have added a clarification above Eq. 11 which, together with the discussion in the paragraph below Eq. 11, should make it more clear to the reader that we consider only one example of the telegraph noise in this section.

Regarding the possible variations in scaling dimension, we have included a discussion of the effect of a deviation from the ideal 1/6 case in the appendix on the anyonic distribution function (now App. C). Since the temperature constriction derived from Eq. 39 is two orders of magnitude below the experimentally relevant temperature, even considerable deviations from the ideal Laughlin case do not break the inequality in Eq. 39. We have included references to the suggested PRL and two additional papers treating this topic. In the main text (Sec. 3), we have specified that we consider the ideal Laughlin edge states.

We agree with the referee that a schematic figure describing the quench protocol is useful. We have included such a figure, which is now Fig. 9.

We thank the referee for pointing out the typo.

Anonymous on 2023-09-21  [id 3998]

(in reply to Ida Nielsen on 2023-08-23 [id 3919])

The authors have made thoughtful changes to the papel that make its impact readily apparent. I recommend publication in its present form.

Anonymous on 2023-09-12  [id 3975]

(in reply to Ida Nielsen on 2023-08-23 [id 3919])

I thank the authors for taking my comments into consideration.
They have addressed them well, and I am happy to reiterate my previous recommendation for publication.

---

## Round 1 · Referee Report · Anonymous (Referee 2) · 2023-6-28

Strengths

1- timely, parafermions are a topic of current experimental interest 2-thorough; the peack of the paper is a protocol for extracting anyonic properties of the quasiparticles from current measurements; sources of noise are identified and modeled in some generality 3-self-contained 4-well written

Weaknesses

Objectively, none. The paper is a typical, high-quality exponent of its subfield. However, my subjective impression reading the paper was sligthly negative because I feel that have read this kind of paper before too many times; the arguments follow closely, perhaps too closely, the pattern set by a myriad of investigations of Majorana fermions. Perhaps the authors should highlight more strongly the most original contributions of the paper.

Report

Recommendation: publish ``as is" or after minor changes.

I have some comments that might suggest some minor modifications. The paper is perfect for people actively working in parafermions, hence already convinced of their value. Readers trying to decide if they would like to join the field actively may find some features of the paper puzzling.

1) The most basic issue is, well, Majorana fermions. Majorana fermions are clearly simpler than parafermions in many respects and yet the experimental search for Majorana fermions has yet to find success. It is tempting to assume that Majorana fermions are somehow a prerequisite for Parafermions but perhaps one can make the case that the two are in fact fairly independent challenges and one is not a prerequisite for the other. What is the authors take on this?

2) While standard, I am puzzled by the choice of tunneling term between the metallic lead and the parafermion mode (\alpha_1). The third power of this mode behaves as a Majorana fermion and is connected to a majorana fermion in the metallic lead. Wouldn't it be more natural to tunnel charged electrons in and out of the metallic lead? The technology for that exists but is rarely used

https://iopscience.iop.org/article/10.1088/1361-648X/aa718f

In short, perhaps it would be useful for the newcomer to discuss more fully the physics of the chosen tunneling term.

  • validity: high
  • significance: good
  • originality: good
  • clarity: high
  • formatting: excellent
  • grammar: excellent

Author:  Ida Nielsen  on 2023-08-23  [id 3920]

(in reply to Report 2 on 2023-06-28)
Category:
answer to question

We thank the referee for the very positive feedback on our work and for the recommendation for publication.

We agree with the referee that signatures of the existence of parafermions (PFs) are indeed independent from the Majorana fermion (MF) experiments. But we respectfully disagree with the referee when they write that “the arguments follow closely, perhaps too closely, the pattern set by a myriad of investigations of Majorana fermions.”: all the features discussed in this paper rely on a separation of the reduced fractional charge $\tilde{q}$ and the overall fermionic parity of the system. The former exists only for PF setups whereas the latter is not conserved. MF setups cannot display any of the signatures we discuss in this work since, in the Majorana scenario, $\tilde{q}$ is not defined.

For example, a strong signature of PFs is the three-level telegraph noise discussed in Sec. 3. This has no counterpart for MFs and cannot be mimicked by trivial states such as Andreev bound states which are often discussed in MF experiments. We have included a comment on this at the end of Sec. 3. Furthermore, the protocol for measurement of associativity rules of the reduced charge number $\tilde{q}$ also has no counterpart in MFs.
Precisely due to this distinction, we do not want to focus on MFs in our work.

Regarding the tunnelling Hamiltonian $H_c$, a crucial reason for the form chosen is the presence of the superconductor (SC) in the PF setup which means that the $H_c$ only conserves charge modulo 2e. In the linked paper by Cobanera, the author uses a Fock PF approach to model tunnelling from electron to fractional modes. In that paper, however, the tunnelling terms are meant to model charge-conserving processes, in which electrons tunnel into FQH edges, for instance, but without a SC. One could consider our $H_c$ as a higher-order process of a Hamiltonian that includes both Cobanera’s charge-conserving tunnelling between normal lead and edge modes, and the superconducting term. Since we are interested in the physics at energies well below the SC gap, our choice depicts in a simpler way the underlying physics. To clarify this better for the reader, we have included a comment about this below Eq. 1.

---

## Round 1 · Referee Report · Anonymous (Referee 3) · 2023-8-21

Strengths

  • Extensive, broad, thorough

Weaknesses

  • Very long and detailed. This is a strength but also a weakness, because the message gets a bit diluted (especially while reading the very long section on the 4-parafermion devices).

Report

Publish in its present form.

The article "Dynamics of parafermionic states in transport measurements" by Nielsen and co-authors is a detailed theoretical investigation of the actual possibilities of detecting Z6 parafermions coupled to electric electrodes.

The article takes into account all possible sources of noise that could appear in these devices and the authors highlight the experimental signatures to be expected and how they could be linked to the presence of parafermions. All different terms appearing in the master equation are thoroughly discussed and introduced. The article is well-organised and although a bit pedantic and technical it discusses in details several physical configurations.

My recommendation is to publish the article as it is.

I have two minor optional questions for the authors.

  • In Fig.2 the authors mark with a red dot the time at which the quantum trajectory projects on one of the three charge sectors. Is it possible to develop an analytical understanding of the pattern of the red dots? Especially in the right panel, some kind of order seems to appear.

  • In the part concerning the coupling to the antidot, I was surprised to read that the antidots are described as two-level systems. I would have expected a 3- or 6-level system, and a model based on Fock parafermions introduced by Ortiz and Cobanera. Could the authors explain their choice?

  • validity: high
  • significance: good
  • originality: good
  • clarity: high
  • formatting: good
  • grammar: perfect

Author:  Ida Nielsen  on 2023-08-23  [id 3921]

(in reply to Report 3 on 2023-08-21)

We wish to thank the referee for the positive report and the recommendation for publication in the present form. We agree with the referee that the report is long and detailed as we wanted to present a coherent and complete story as opposed to submitting two separate manuscripts with similar methods.

Regarding the two questions from the referee:

The order appearing in Fig. 2(b) can indeed (in part) be explained and analysed by the continuous part of the wave function evolution, i.e. the evolution operator $\mathcal{U}$ in Eq. (29). The component containing the product of jump operators leads to a wave function decay that depends on the charge sector (the value of $q$). This is explained in more detail in the second half of appendix B (previously appendix A), and in Fig. 11(a) (previously 10(a)) an analytic estimate is plotted in white on top of Fig. 2(b).

The approximation of the antidot as a two-level system is the simplest case where two many-body states differing in charge by e/3 are almost degenerate with a small splitting $\delta_{ad}$. This is likely to represent the physics of small antidots with sizeable effective charging energies, close to one of their charge degeneracy points. For larger antidots with lower charging energies, three or more states with different charges could have comparable energies and thereby be relevant to consider for poisoning. This is briefly discussed in the conclusion in Sec. 6, and we added a further comment about the two-level approximation at the beginning of Sec. 4.

Anonymous on 2023-08-26  [id 3930]

(in reply to Ida Nielsen on 2023-08-23 [id 3921])

Dear Editor,

I would like to thank the authors for their reply. I feel that all criticisms have been answered and that the article can be published in its present form.

---

## Editorial Decision

resubmitted